

# Insights into the diurnal cycle of global Earth outgoing radiation using a numerical weather prediction model

Jake J. Gristey[1], J. Christine Chiu[1,2], Robert J. Gurney[1], Cyril J. Morcrette[3], Peter G. Hill[1], Jacqueline E. Russell[4], Helen E. Brindley[4,5]

[1]Department of Meteorology, University of Reading, UK
[2]Department of Atmospheric Science, Colorado State University, USA
[3]Met Office, Exeter, UK
[4]Department of Physics, Imperial College London, UK
[5]NERC National Centre for Earth Observation, UK

*Correspondence to:* Jake J. Gristey (J.Gristey@pgr.reading.ac.uk)

**Abstract.** A globally-complete, high-temporal resolution and multiple-variable approach is employed to analyse the diurnal cycle of Earth's outgoing energy flows. This is made possible via the use of Met Office model output for September 2010 that is assessed alongside regional satellite observations throughout. Principal component analysis applied to the longwave component of modelled outgoing radiation reveals dominant diurnal patterns related to land surface heating and convective cloud development, respectively explaining 68.5 % and 16.0 % of the variance at the global scale. The total variance explained by these first two patterns is markedly less than previous regional estimates from observations, and this analysis suggests that around half of the difference relates to the lack of global coverage in the observations. The first pattern is strongly and simultaneously coupled to the land surface temperature diurnal variations. The second pattern is strongly coupled to the cloud water content and height diurnal variations, but lags the cloud variations by several hours. We suggest that the mechanism controlling the delay is a moistening of the upper troposphere due to the evaporation of anvil cloud. The shortwave component of modelled outgoing radiation, analysed in terms of albedo, exhibits a very dominant pattern explaining 88.4 % of the variance that is related to the angle of incoming solar radiation, and a second pattern explaining 6.7 % of the variance that is related to compensating effects from convective cloud development and marine stratocumulus cloud dissipation. Similar patterns are found to exist in regional satellite observations. The first pattern is controlled by changes in surface and cloud albedo, and Rayleigh and aerosol scattering. The second pattern is strongly coupled to the diurnal variations in both cloud water content and height in convective regions but only cloud water content in marine stratocumulus regions, with substantially shorter lag times compared with the longwave counterpart. This indicates that the shortwave radiation response to diurnal cloud development and dissipation is more rapid, which is found to be robust in the regional satellite observations. These global, diurnal radiation patterns and their coupling with other geophysical variables demonstrate the process level understanding that can be gained using this approach and highlight a need for global, diurnal observing systems for Earth outgoing radiation in the future.



## 1 Introduction

Solar radiation entering the top of the atmosphere (TOA) is the primary energy source for atmospheric processes on Earth. Around a third of this radiation is returned directly to space as reflected solar radiation (RSR). The remainder is absorbed by
the atmosphere and surface, acting to constantly heat the Earth. The Earth is, in unison, constantly losing heat energy to space in the form of outgoing longwave radiation (OLR). The RSR and OLR, collectively referred to as Earth outgoing radiation (EOR), approximately balance the incoming solar radiation when globally and annually averaged, maintaining a state of equilibrium in the global energy budget (e.g., Trenberth, 2009; Stephens et al., 2012; Wild et al., 2015). Understanding the physical nature and influences on the processes that determine the variability in the global energy budget lies at the heart of
climate science research.

While the incoming solar radiation is relatively stable, predictable and observed with high accuracy (e.g., Kopp and Lean, 2011), EOR is dynamic by nature and therefore inherently more difficult both to observe and to understand. This is perhaps manifested most clearly in the strong diurnal signatures that EOR exhibits, a direct result of the rapidly evolving scene from which the radiation originates. Diurnal variability in the Earth system that defines such signatures has been studied extensively
(e.g., Nitta and Sekine, 1994; Webster at al., 1996; Soden, 2000; Yang and Slingo, 2001; Wood et al., 2002; Nesbitt and Zipser, 2003; Taylor, 2012). However, discrepancies persist when comparing the diurnal cycles in observations and models (e.g., Betts and Jakob, 2002; Dai and Trenberth, 2004; Slingo et al., 2004; Tian et al., 2004; Itterly and Taylor, 2014). These discrepancies highlight a lack of understanding, yet it is essential we can correctly represent diurnal variability since it constitutes a fundamental forcing cycle for our weather and climate.

Previous attempts to identify patterns of diurnal variability in EOR have made use of principal component analysis (PCA). For example, Smith and Rutan (2003) performed PCA on seasonally averaged OLR observations bounded by 55°N to 55°S from the scanning radiometer aboard the Earth Radiation Budget Satellite (ERBS) (Harrison et al., 1983; Barkstrom, 1984). They found dominant patterns that appeared to be related to heating of the surface and lead-lag effects from the development of cloud, noting that the patterns over ocean and land explain significantly different amounts of variance. Comer et al. (2007)
applied a similar method to OLR observations from the Geostationary Earth Radiation Budget (GERB) instrument (Harries et al., 2005) but, instead of separating land and ocean, chose to consider the domain as a whole. The dominant patterns in the GERB observations were similar to those found by Smith and Rutan (2003) but the orography of the land was used to support the explanation of the patterns, illustrating the value of additional information for understanding the physical processes involved. The dominant OLR patterns of variability revealed by PCA also provide a useful tool for comparing and evaluating
the diurnal cycle of OLR in climate models (Smith et al., 2008).

By contrast with OLR, the diurnal cycle of RSR has received less attention. This is likely due to its non-continuous nature and relatively complex variations. To our knowledge only Rutan et al. (2014) have considered RSR by using observations from the ERBS, similar to Smith and Rutan (2003), to perform PCA on the diurnal cycle of TOA albedo. Interestingly, they



found the diurnal cycle of TOA albedo is primarily driven by a dependence on solar zenith angle (SZA) and that any other
signals are an order of magnitude smaller.

The aforementioned studies represent the forefront of our knowledge regarding the dominant patterns of diurnal variability in EOR. However, none of the datasets used in those studies permit the global coverage required for relating the revealed patterns back to the global energy budget, nor do they use variations in other geophysical data to support physical interpretation of the patterns. A numerical weather prediction (NWP) model provides a unique tool for achieving these criteria. Clearly, care
must be taken to analyse the model data in line with its ability to reproduce real world processes, but the wealth and variety of data available undoubtedly enables a deeper understanding at the process level. It is intended that any process level understanding obtained from analysis of NWP model output will help to formulate hypotheses that can be tested later with observations.

Here we perform PCA on global output from the Met Office NWP model. The dominant patterns of variability that this
reveals will be supported by satellite observations and radiative transfer calculations where possible. Section 2 outlines details of the model run and supporting satellite datasets. Section 3 describes the method of identifying and interpreting patterns of diurnal variability. Section 4 reports our findings that, crucially, take three distinct steps forward. In Sect. 4.1, we examine the dominant patterns of diurnal variability in OLR at a fully global scale for the first time, required for relating the dominant patterns to the global energy budget. In Sect. 4.2, we examine the dominant patterns in the diurnal variability of TOA albedo,
using the surface and cloud free fluxes combined with radiative transfer calculations to reveal the processes contributing to the patterns. In Sect. 4.3, the patterns of EOR variability are coupled with variability in other relevant geophysical variables to aid their physical interpretation. Section 5 summarizes the results and conclusions are drawn.

## 2 Data

### 2.1 Global model output

The main data used in this analysis are synthetic global EOR fields generated using the Met Office Unified Model in its global NWP configuration. We used the Global Atmosphere 6.0 (GA6) and Global Land 6.0 (GL6) components, described by Walters et al. (2017), with sea surface temperatures and sea ice prescribed from the Operational Sea-surface Temperature and sea Ice Analysis (Donlon et al., 2012). Operationally, for reasons of computational expense, full radiation calculations are not done every time step (Manners et al., 2009). In GA6, the full radiation calculations are done every hour, with an update to
represent the changing cloud fields every 12-minute time step (Walters et al., 2017; Manners et al., 2009). In this simulation however, the full radiation scheme, based on Edwards and Slingo (1996), was called on every model time step to better represent the evolution of EOR. The model was run with this setup for each day from an operational 0000 Z analysis.

The data are provided for each day in September 2010. The year of 2010 was chosen arbitrarily, but the month of September was selected specifically due to the timing of the equinox. At the equinox the day length is approximately constant at all
locations on Earth, so the months containing the equinoxes are the only times during the year that a consistent diurnal cycle



can be assessed globally. In particular, RSR only has a signature during the daylight hours so away from the equinoxes the analysis would be fundamentally limited in one of the hemispheres. While dominant patterns in the diurnal cycle of EOR do exhibit spatial variations between seasons, the relative importance of physical processes that control the dominant patterns typically remain robust throughout the year (Smith and Rutan, 2003; Rutan et al., 2014). This allows insight to be gained for 100 the entire annual cycle, at least in a qualitative sense, by just considering this unique situation.

The data are provided with a 12-minute temporal resolution (i.e. at every model time step) on the N320 grid, giving a spatial resolution of approximately 40 km in mid-latitudes. These temporal and spatial resolutions are selected to retain all relevant information while avoiding data redundancy. This is based on initial experiments where we artificially reduced the temporal/spatial resolutions in one day of very high resolution (5-minute/~17 km in mid-latitudes) global EOR fields, and 105 found that the dominant patterns in the data (see Sect. 3.2) are well retained at a resolution of 15 minutes/~50 km in mid-latitudes.

When analysing the RSR we work with the TOA albedo, similar to Rutan et al. (2014), calculated as the division of outgoing by incoming TOA solar irradiance. This normalization removes the variability associated with the amount of incoming solar radiation that would otherwise dominate the diurnal cycle, but is not of interest here. For September 2010 it is possible to 110 define the TOA albedo from 0700–1700 local solar time and from 61.5°N to 61.5°S, encompassing over 94 % of the total incoming SW irradiance entering the Earth system.

As well as the OLR and TOA albedo, a host of other geophysical variables were simultaneously output from the model to aid the physical interpretation of the EOR diurnal cycles. The additional variables included in this study are the equivalent surface and clear-sky radiation fluxes, surface temperature, cloud liquid water path (LWP), cloud ice water path (IWP) and 115 cloud top height (CTH).

## 2.2 Supporting satellite datasets

Several observational datasets of EOR currently exist that are derived from various satellite instruments. Global EOR observations, such as those from the Clouds and the Earth's Radiant Energy System (CERES) instrument (Wielicki et al., 1996), provide complete coverage but are not used in this study mainly due to their lack of diurnal sampling from low-Earth 120 sun-synchronous orbits. Substantial efforts have been applied to interpolate between the diurnal gaps in CERES sampling (Doelling et al., 2013; 2016) but these products do not match the high temporal resolution of the model data required for thorough investigations of the diurnal cycle. Observations from the Scanner for Radiation Budget (ScaRaB) instrument are capable of capturing long-term averaged diurnal variability due to the drifting orbit of the Megha-Tropiques satellite (Viollier and Raberanto, 2010), but are limited to the inner tropics due to the very low-inclination of the orbit and are therefore also not 125 appropriate. GERB observations however, made from the unique vantage point of geostationary orbit, provide EOR at high temporal resolution over a large region including Africa, Europe and their surrounding waters and are therefore much better suited to this study. An added advantage is that simultaneous retrievals of cloud properties are available from the Spinning



Enhanced Visible and InfraRed Imager (SEVIRI) instrument (Schmid, 2000). We therefore choose to use the GERB and SEVIRI observations to support our model analysis.

Specifically, we make use of OLR and TOA albedo observations from GERB 2 (GERB Edition 1 High Resolution (HR) product with "SW combined adjustment" applied) and CTH observations from SEVIRI (Climate Monitoring Satellite Applications Facility (CMSAF) Cloud Property DAtAset using SEVIRI (CLAAS) Edition 2 product; Benas et al. (2017)). We do not include the SEVIRI LWP and IWP products because the retrieval method, which assumes that the cloud phase is the same as cloud top for the whole column, leads to unphysical diurnal variability during convective cloud development, which

turns out to be an important process in the diurnal cycles of both OLR and TOA albedo as will be shown in Sect. 4. To avoid missing data in the Southern Hemisphere and high uncertainty data near the edge of the field-of-view (FOV) we use data north of 20° S and with a viewing zenith angle of less than 70°, respectively. Unfortunately, the time window of the model and observation data cannot be matched because full diurnal GERB observations are not available close to the equinoxes due to potential instrument damage. Instead we use observations from July 2006. This month accommodates large solar insolation

over the Northern Hemisphere land mass in the GERB FOV that should amplify any diurnal signatures in these regions, and was also the subject of the Comer et al. (2007) study.

Note that the longitudinal coverage of GERB has recently extended to include the Indian Ocean (Dewitte et al., 2017), but the coverage remains well short of global. This lack of global coverage removes the opportunity to investigate processes across regions that is afforded by the model data, but at least allows us to evaluate our model results over one portion of the globe.

The potential for global diurnal sampling of EOR from a single observing system has recently been highlighted via the use of a constellation of small satellites (Gristey et al., 2017) but, for now at least, observations required to fully resolve the diurnal cycle in global EOR do not exist.

## 3 Method

### 3.1 Pre-processing

Before performing PCA, we must ensure that the data fields are in an appropriate format for extracting patterns of diurnal variability. This involves conversion of the diurnal time coordinate, creation of an average diurnal cycle and a correction to account for changes in grid resolution, implemented as follows.

First, all data fields are transformed from UTC to local solar time. This is required such that all spatial locations correspond to the same part of the diurnal cycle. To achieve this transform, we note that each longitude column in UTC represents a single

local solar time. We then select the longitude columns from each UTC map that correspond to the same local solar time and combine them to generate a new set of maps that are now a function of local solar time.

Next, we calculate the monthly average diurnal cycle for each data field by simply averaging the local solar time maps from each day in the month. Since the variations on any given day consist of not only diurnal variations, resulting from the



periodic forcing, but also transient weather variations, which are not diurnally forced, performing this monthly averaging helps
to reduce the noise from weather events and extract the signal from the diurnal variations of interest.

Lastly, since the data are on equal latitude-longitude grids, we apply a latitude correction by multiplying each grid point by the square root of the cosine of its latitude. This avoids spurious poleward enhancement of variability due to the changes in grid spacing (e.g., Wallace et al., 1992; Comer et al., 2007; Bakalian et al., 2010).

Note that we do not separate data over ocean and land before performing PCA. This is because we intend to reveal global
patterns and their relative importance across all regions. Comer et al. (2007) showed that the behaviour of the system can be captured well by considering the diurnal cycles over ocean and land simultaneously.

### 3.2 Extracting dominant patterns of EOR diurnal variability

PCA applied to the local solar time, monthly averaged and latitude corrected fields of OLR and TOA albedo extracts empirical orthogonal functions (EOFs) and principal components (PCs) that reveal spatial and temporal patterns in the data,
respectively. The first PC describes the maximum possible variance, and each subsequent PC describes the maximum possible variance remaining once the preceding PCs have been removed. There are several approaches to achieve PCA. The approach used is this study is outlined below.

First, we generate a data matrix, $F$, containing the spatial-temporal data to be used as input for the PCA. The matrix $F$ has $t$ rows and $s$ columns, where $t$ is the number of time steps in the diurnal cycle and $s$ is the total number of spatial grid points.
In other words, each row of $F$ consists of a flattened map of the data field at a given local solar time, and each column represents a time series at a given location. Additionally, the mean is removed from each column of $F$ to give an anomaly time series.

In a standard PCA one would next form the large covariance matrix, $R$, of $F$ given by

$$R = F^T F,\tag{1}$$

and perform an eigenvalue decomposition on $R$ to obtain the EOFs. However, in this application $F$ is very non-square (the
spatial dimension is much greater than the temporal dimension), which would result in a very large $s \times s$ covariance matrix from Eq. (1) and an expensive eigenvalue decomposition. To reduce computational expense, we follow the equivalent method to obtain the leading EOFs and PCs by forming the smaller $t \times t$ covariance matrix, $R^*$, given by

$$R^* = FF^T.\tag{2}$$

The eigenvalue problem for the small covariance matrix, $R^*$, in Eq. (2) is formulated as

$$R^* C^* = C^* \lambda',\tag{3}$$

where $C^*$ is a $t \times t$ matrix with columns comprising the eigenvectors of $R^*$; and $\lambda'$ is a $t \times t$ diagonal matrix containing the corresponding eigenvalues in descending order. For convenience, the diagonal elements of $\lambda'$ are placed into $\lambda$, a row vector of length $t$.



The eigenvalues, $\boldsymbol{\lambda}$, from Eq. (3) are also the leading eigenvalues of the large covariance matrix, $\boldsymbol{R}$, in Eq. (1). However, the leading eigenvectors of $\boldsymbol{R}$ are not $\boldsymbol{C}^*$, but are represented by columns in a $s \times t$ matrix $\boldsymbol{C}$. These column vectors, $\boldsymbol{C}_j$, are calculated as

$$\boldsymbol{C}_j = (\boldsymbol{F}^T \boldsymbol{C}^*)_j / \sqrt{\lambda_j}. \tag{4}$$

A proof of this relationship is provided by Bjornsson and Venegas (1997). The column vectors $\boldsymbol{C}_j$ are the EOFs that we seek. For illustrative purposes, we scale each EOF such that the maximum absolute value is ten.

The corresponding PC is calculated by projecting the original data matrix, $\boldsymbol{F}$, on to the EOF, $\boldsymbol{C}_j$, in Eq. (4) as

$$\boldsymbol{A}_j = \boldsymbol{F}\boldsymbol{C}_j, \tag{5}$$

where $\boldsymbol{A}_j$, a column vector of length $t$, is the PC that we seek.

The percentage variance, $\tau_j$, explained by the EOF/ PC pair from Eq. (4) and Eq. (5) is

$$\tau_j = (\lambda_j / \sum_{n=1}^{t} \lambda_n) \times 100. \tag{6}$$

### 3.3 Coupling dominant patterns of diurnal variability

To aid physical interpretation of the leading EOR EOFs, the corresponding PCs, and the percentage variance they explain, respectively calculated from Eq. (4)–(6), we also investigate their extent of coupling with the variability in other geophysical variables. Coupled PCA patterns between multiple variables have been widely examined in the weather and climate sciences (e.g., Kutzbach, 1967; Wallace et al., 1992; Deser and Blackmon, 1993; Zhang and Liu, 1999) but this additional step has not been applied in previous PCA studies of EOR.

A comprehensive overview of the advantages and disadvantages of common techniques used to identify coupled patterns is given by Bretherton et al. (1992). Here, we are interested in the relationship between a selected pattern of variability in EOR and all of the variability in another variable, which is well suited to an analysis technique previously referred to as single-field PCA (e.g., Wallace et al., 1992). In our application, this will involve studying the correlations between a PC in either OLR or TOA albedo with the diurnal cycle of another variable that we expect to be related to the PC. These correlations are illustrated as heterogeneous correlation maps, which reveal the spatial distributions of where the selected EOR PC has the highest correlations with the diurnal variability in the other variable.

Before generating the heterogeneous correlation maps, we first perform a cross-correlation between the selected EOR PC and the leading PC of the other variable to identify any lag between the patterns. Both PCs represent global time series with the rationale that the radiation PC is dominated by a certain process, and the other PC exhibits variability directly related to that process. The cross-correlation is achieved here by calculating a set of Pearson correlation coefficients between the PC of the other variable, which remains fixed in time, and the EOR PC, that is shifted by one time step at a time throughout the entire diurnal cycle. For the TOA albedo, the correlation coefficients are calculated for the time window over which it is defined. From this cross-correlation we can extract the maximum correlation coefficient magnitude, giving an indication of the strength of coupling, and the lag time at which it occurs, giving an indication of how out of phase the patterns are. We define the lag





time to be positive when the PC of EOR follows the PC of the other variable (e.g. a change in OLR occurs after the development of LWP). The lag time is then removed before calculating the heterogeneous correlation maps.

The lag times themselves also provide insight into processes and their evolution. We therefore calculate the lag times between various radiation and cloud variables in both the model and GERB/SEVIRI observations. Since the observational data

are provided on an irregular grid, we linearly interpolate the observational data onto the same grid as the model data in order to perform the local solar time conversion (see Sect. 3.1).

## 4 Results

### 4.1 Dominant patterns of diurnal variability in modelled global OLR

The first EOF of the global OLR diurnal cycle (Fig. 1a) reveals positive weights, indicating a consistent sign in the diurnal

variations, over land surfaces that are largest in arid regions such as the Sahara Desert, Atacama Desert and Arabian Peninsula. The corresponding PC (Fig. 1c) reaches maximum amplitude just after local midday and minimum amplitude overnight. This spatio-temporal pattern is consistent with that expected from solar heating of the land surface and accounts for 68.5 % of the global diurnal variance. Although it is primarily the surface that is being heated, it should be noted that transmittance of longwave radiation back through the atmosphere is often low, typically less than 10 % at the global scale (Costa and Shine,

2012). A large fraction of the variation in OLR reaching the top of the atmosphere as a result of solar heating of the land surface is therefore likely to be due to radiation that has been absorbed and re-emitted by the atmosphere.

The second EOF (Fig. 1b) contains consistent features across many different regions, but the features themselves are small in spatial extent and therefore difficult to interpret at the global scale. When examining the Maritime Continent region as an example (Fig. 2), we find positive weights over the islands that are enhanced along the coastlines, and negative weights just

offshore. Similar patterns are seen in other coastal regions in the tropics. The corresponding PC (Fig. 1d) shows that these patterns are at a minimum in the late afternoon, and a maximum in the early morning. This spatio-temporal pattern, accounting for 16.0 % of the global diurnal variance, is consistent with the OLR signature from the cold tops of deep convective clouds that develop over land during the late afternoon, and over the oceans in the early morning. The unique topography of this region permits strong sea breezes (Qian, 2008) explaining the enhancement along the coastlines. Note that in the studies by

Smith and Rutan (2003) and Rutan et al. (2014) coastal data are omitted. The spatial patterns of OLR in this region also match surprisingly well with retrieved rainfall at different times during the diurnal cycle, as presented by Love et al. (2011) using observations from the Tropical Rainfall Measuring Mission (TRMM). However, the timing of the minimum modelled OLR signal is substantially earlier than the peak in TRMM retrieved rainfall, consistent with well-documented model biases in the timing of convection (e.g. Yang and Slingo, 2001).

Both of the dominant EOFs and PCs of OLR diurnal variability in Fig. 1 are, reassuringly, similar to those identified with GERB (Comer et al., 2007) and ERBS (Smith and Rutan, 2003) observations, despite the different regions and time periods considered. However, what is markedly different is the percentage variance that these patterns account for. Comer et al. (2007)





considered the domain of analysis as a whole rather than separating land and ocean, facilitating a direct comparison with our results. The variances explained by the dominant patterns in their study were 82.3 % and 12.8 %, respectively. To first order, this suggests that their results exhibit a higher relative contribution from surface heating to the OLR diurnal variability and a lower relative contribution from convective processes, although there may also be an influence from the fact that the actual diurnal variations in some regions can be better explained by a contribution from both dominant patterns for reasons such as surface thermal lag (Futyan, and Russell, 2005). This only appears to be the case for a select few regions including the Tibetan Plateau and parts of Southern Africa. However, the total variance explained by their first two patterns is higher at 95.1 % compared with 84.5 % in our results. These differences could be a result of the different time periods and spatial regions considered, or model-observation discrepancies such as the fixed sea surface temperatures in the model. To isolate the influence from different spatial regions, we repeated our analysis using the model data sub-sampled over the GERB FOV (not shown), and found that the total variance explained by the first two patterns increases to 89.6 %, indicating that around half of the difference is due to the disproportionately high fraction of land mass within the GERB FOV. This is because the first two dominant patterns of OLR diurnal variability appear to be driven, directly and indirectly, by solar heating of land mass. Interestingly, this suggests that the relative importance of diurnal processes acting within the GERB FOV, the only portion of the Earth that we currently make well resolved diurnal observations of EOR, may not be representative of the global OLR diurnal cycle.

## 4.2 Dominant patterns of diurnal variability in TOA albedo

### 4.2.1 From model output

PCA is repeated for the TOA albedo diurnal cycle. The dominant pattern of variability, explaining 88.4 % of the total variance, consists of an EOF (Fig. 3a) with positive weights everywhere, and a diurnally symmetric PC (Fig. 3c) that follows the inverse timing of incoming shortwave irradiance. The dominance of this leading spatio-temporal pattern, despite being consistent with observations from the ERBS (Rutan et al. 2014), is somewhat surprising given that the TOA albedo is a quantity normalized by the amount of incoming solar radiation. This dominance indicates a strong dependence of the TOA albedo on the SZA itself that has been well documented in empirically-based angular distribution models (Loeb et al., 2003; Loeb et al., 2005; Su et al., 2015), but warrants further investigation into the physical processes at play.

The first PC in Fig. 3c has a U-shape feature, representing a dependence on $1/\mu_0$, where $\mu_0$ is the cosine of SZA. To illustrate how the cloud-free atmosphere contributes to the shape, Fig. 4 shows TOA albedos from offline radiative transfer simulations under various simplified situations. For a typical example of an aerosol-free atmosphere, we see that Rayleigh scattering dominates and that atmospheric absorption is only able to counteract this dependence when the Rayleigh scattering is scaled down to around 10 % of its original value (Fig. 4a). Adding a moderate amount of aerosol into the simulations (Fig. 4b), we find that the U-shape is retained, but scaled to a different magnitude. In fact, this U-shape is not limited to certain atmosphere setups or aerosol types because, in low optical depth atmospheres, different optical depths, single scattering



albedos and asymmetry parameters only provide a scaling of the shape. In other words, the reflectance function of the atmosphere under a single-scattering approximation always retains a dependence on $1/\mu_0$ since this is the factor by which the path length increases and heightens the chance of a scattering evert occurring. As a result, the first EOF (Fig. 3a) exhibits positive weights in many different predominantly cloud-free regions, such as the global deserts.

The influence of the surface and cloud is also clearly evident in the first EOF. There are generally larger weights over the
ocean than the land, and the largest weights occur in regions of persistent cloud (e.g. marine stratocumulus regions, and the inter-tropical convergence zone (ITCZ)). The larger diurnal variations over the oceans can be seen by comparing the global-mean diurnal cycle of TOA albedo separated over land and ocean explicitly (Fig. 5a). The reason for these differences is revealed by examining the diurnal cycle in the albedo defined at the surface (Fig. 5b), where we find the albedo over land surfaces is larger and diurnally constant. The erosion of the U-shape by brighter surfaces can be seen in Fig. 4b, and the SZA
dependence of the surface albedo itself follows directly from the setup of surface albedo in the model, which is Lambertian over land, but uses a modified version of the parametrization from Barker and Li (1995) over ocean. Similarly, the larger diurnal variations in the presence of cloud can be seen by comparing the global-mean diurnal cycles of all-sky (Fig. 5a) and clear-sky (Fig. 5c) TOA albedo. The differences are particularly evident over land where the diurnal range in global mean albedo reduces from 0.11 in the all-sky field to 0.07 in the clear-sky field. This is consistent with the sharp contrast in the EOF
over land (Fig. 3a) between predominantly cloudy regions, such as along the ITCZ over central Africa, and predominantly clear-sky regions, such as immediately north of the ITCZ over Africa. Over both land and ocean surfaces, cloud introduces a more rapid change in the TOA albedo close to midday when the incoming solar radiation is most intense.

The second EOF of the TOA albedo diurnal cycle (Fig. 3b) contains many smaller scale features similar to that of the second EOF for OLR. In fact, zooming in to the Maritime Continent region again (Fig. 6) reveals very similar patterns. The
corresponding PC (Fig. 3d) however, is reversed in sign when compared with the second PC for OLR. This is consistent with the enhanced reflection from convective clouds that develop over land during the late afternoon, and over the oceans in the early morning. This acts to skew the TOA albedo diurnal cycle to earlier in the day over land (minimum around 11:20 local solar time) and later in the day over the oceans (minimum around 12:10), which is evident in Fig. 5a. This spatio-temporal pattern explains just 6.7 % of the total variance.

Although the patterns in the second EOFs of TOA albedo and OLR are remarkably similar in the Maritime Continent region, there are obvious differences in other regions. In particular, the marine stratocumulus regions located to the west of continental land masses exhibit negative weights in the TOA albedo EOF that do not appear in the OLR EOF. This signal appears to be related to the diurnal development and dissipation of marine stratocumulus clouds themselves, and is not apparent in the OLR since these variations occur close to the surface. The diurnal cycle of these clouds has been well characterized by
ship track observations (Burleyson et al., 2013) and more extensive field campaigns (Boutle and Abel, 2012) as having a maximum thickness overnight/during the morning and a minimum thickness during the afternoon/evening induced by solar absorption of the cloud layer, which appears to be captured by the model. The fact that the diurnal cycles of convective cloud (e.g. in the Maritime Continent region) and marine stratocumulus cloud (e.g. to the west of continental land masses) are present





in the same pattern of variability is noteworthy in itself. Their opposite sign suggests that they are leading to compensating

effects: the enhanced reflection from the development of convective cloud in the afternoon is compensated by the reduced reflection from dissipating marine stratocumulus cloud.

### 4.2.2 From GERB observations

Finally, we present a PCA of TOA albedo using GERB observations (Fig. 7) and compare it with results from the model data, noting that the modelled patterns are similar when sub-sampled over the GERB FOV (not shown). The leading pattern

of variability remains very dominant, explaining 79.5 % of the variance. The first EOF (Fig. 7a) matches the patterns in the model data well, repeating the larger positive weights over the ocean, the South-East Atlantic marine stratocumulus region and equatorial Africa. The northward migration of the ITCZ between September (model fields) and July (GERB observations) is evident over Africa. The first PC (Fig. 7c) also matches the models diurnally symmetric timing of this pattern associated with the SZA dependence.

The second pattern of variability, explaining 15.1 % of the variance, consists of an EOF (Fig. 7b) that contains similar features to those in the second EOF of OLR in the study by Comer et al. (2007) attributed to convective cloud development. However, just like the equivalent model EOF, this EOF also contains negative weights around the west coast of Southern and Central Africa and the South-East Atlantic related to the diurnal cycle of marine stratocumulus cloud. These patterns provide observational support that the compensating influences of convective and marine boundary layer cloud evolution on the TOA

albedo are robust. The positive peak of the second PC (Fig. 7d) however, is shifted to slightly later in the day compared with the model results. The later timing of peak convection in reality compared to the model could be what is pulling the observational PC to later in the day, but the marine stratocumulus variations appear to follow the shift as well, suggesting that the stratocumulus could also be breaking up too early in the model. Unrealistic breaking up of marine stratocumulus in the Met Office model has been previously documented by Allan et al. (2007). One consequence of this shift is that the peak in the

second PC of TOA albedo appears to fall outside the 0700–1700 time window over which the albedo is defined in the observations. In summary, the processes controlling the dominant patterns of variability in the diurnal cycle of TOA albedo appear to be consistent between the model and GERB observations.

The presence of distinctly different cloud variations in the same EOF is insightful in this case, but equally highlights a weakness in the PCA method for identifying unique physical processes. That is to say, if two or more physically independent

processes are occurring approximately in phase, or indeed with opposing phase as is the case here, they become statistically linearly related and will be incorporated into the same pattern of variability. The unique identification of such processes then relies on revealing the spatial and temporal coupling of the dominant patterns with other relevant geophysical variables, as examined next.



### 4.3 Coupled patterns of diurnal variability in EOR and other geophysical variables

The physical interpretation of the dominant pattern of variability in modelled TOA albedo was supported by additional surface albedo and clear-sky TOA albedo data fields as well as offline radiative transfer calculations. Thus far however, the interpretation of the other modelled patterns presented in Sect. 4.1 and Sect. 4.2 (i.e. the two leading OLR patterns and the second TOA albedo pattern) has been limited to analysis of the EOFs and PCs alone. We now build a stronger argument for relating those statistical patterns to physical behaviour by assessing their extent of coupling with diurnal variability in other model variables directly related to the previously suggested behaviour.

#### 4.3.1 OLR

The cross-correlation of the first PC of OLR and the first PC of the modelled surface temperature field reveals a very high and near-simultaneous correlation (Table 1, row 1), demonstrating that the temporal structures of these patterns are highly coupled. The reported lag time of −0.2 hours represents a single 12-minute model time step and the correlation is almost identical at no lag, so the lag of −0.2 hours rather than 0 hours likely has no physical relevance and the patterns can be considered to be simultaneously varying. Spatially, the first PC of OLR is highly correlated with the diurnal cycle of surface temperature at each grid point over land (Fig. 8a), indicating the spatial patterns are also highly coupled. Near the poles the diurnal cycle is poorly defined leading to the spurious negative correlations. Over ocean there is no correlation, because the model sea surface temperatures are prescribed from a fixed daily field and do not exhibit diurnal variability. This could be addressed in future work by considering a configuration of the model that is coupled to the ocean. If this was done we would expect some positive correlation over the oceans due to solar heating of the ocean surface, but the amplitude of the diurnal surface temperature change would be much weaker than that over land.

The cross-correlation of the second PC of OLR with the first PCs of modelled variables that are related to convective cloud development (LWP, IWP and CTH) reveals very high correlations but with substantial lag times (Table 1, rows 2–4). A lag between these variables is expected during convective cloud development, and the order in which the lags occur is consistent with the lifecycle of a convective system. As convection initiates, water will begin to condense and cloud will develop at warm lower levels causing the LWP to build first and the longest lag time. Once the convection breaks through the freezing level, further cloud development will mostly consist of ice crystals and the IWP will build leading to a relatively shorter lag time. All the while, the vertical extent of the cloud is increasing and, as the convective system matures and produces an anvil, the CTH will reach a maximum providing the shortest lag time. At this stage, as the convection dies and the CTH begins to reduce, one may expect the OLR to respond immediately but, curiously, a 3-hour lag remains between the maximum correlation of OLR PC2 and CTH PC1.

A possible explanation for this remaining lag is provided by considering the changes in the environment of the upper troposphere after the convection dissipates. As the anvil cloud horizontally entrains into surrounding clear-sky regions it will evaporate, leading to an increase in upper tropospheric humidity (UTH). Using one year of longwave water vapour (6.7 μm)




and window (11 μm) channel radiances from multiple geostationary satellites spanning global longitude, Tian et al. (2004) showed that deep convection in the tropics acts to moisten the upper troposphere via the evaporation of anvil clouds generated by deep convection. This increase in UTH can be prevalent over large spatial extents and will delay the increase in OLR after

the convective cloud has dissipated due to continued absorption of the more intense radiation originating from warmer, lower altitudes. The radiative heating that this provides leads to an increase in atmospheric stability and limits further cloud development providing an important radiative-convective feedback mechanism for the diurnal cycle (Stephens et al., 2008). The study by Tian et al. (2004) suggested a lag of approximately 6 hours between high cloud cover and UTH. In a similar but more spatially- and temporally-limited analysis, Soden (2000) suggested a lag time of approximately 2 hours. In fact, the lag

can be quite uncertain as it depends on the initial state of the atmosphere and spatial scale of convection (Ingram, 2015). The 3 hour lag found here falls between these values and suggests that diurnal variations in OLR due to convective activity may remain tied to the UTH even when the convective cloud itself has dissipated.

To assess the spatial correlations we return to the Maritime Continent region where we know there are strong diurnal cycles in convective activity. The second PC of OLR correlated with the diurnal cycle of LWP at each grid point (Fig. 8b) shows the

highest correlations in the same regions as the largest weights in the second OLR EOF. This indicates that this pattern of OLR variability is highly coupled to diurnal cloud development in these regions, as expected. Similar patterns are seen for IWP and CTH.

### 4.3.2 TOA albedo

The cross-correlation of the second PC of TOA albedo with the first PCs of LWP, IWP and CTH reveals systematically

higher correlations than the corresponding OLR correlations (Table 2). The order of the lag times amongst the cloud variables is maintained, but the lag times are shorter and only a one hour lag remains between the TOA albedo and CTH. Unlike the OLR, the TOA albedo will not continue to respond in a similar way to the cloudy atmosphere once the cloud evaporates and the UTH increases. In fact, the opposite will occur as more solar radiation is absorbed in the humid environment. Remember that the second TOA albedo PC is also controlled by marine stratocumulus cloud that will not moisten the upper troposphere,

and may reduce the time lag between the variations in TOA albedo and CTH. The implication of this differing radiation response is that the diurnal changes in TOA albedo due to cloud development and dissipation are sharper and more immediate. Conversely, the OLR response is spread over a larger time and occurs later.

Similar to the second PC of OLR, the spatial correlation of the second PC of TOA albedo with the diurnal cycle of LWP at each grid point in the Maritime Continent region (Fig. 8c) shows the highest correlations in the same regions as the largest

weights in the second TOA albedo EOF. Again, similar patterns are seen for the IWP and CTH in this region. In marine stratocumulus regions however (not shown), the correlations are high for LWP but not for the other variables, demonstrating the value of assessing the extent of coupling with different data fields to identify unique physical processes. A schematic diagram summarizing the dominant processes controlling the OLR and TOA albedo, and their relation to other variables, is provided in Fig. 9.



### 4.3.3 Lag times in GERB and SEVIRI observations

We finally present the equivalent correlations and lag times in GERB observations of OLR and TOA albedo and SEVIRI observations of CTH for July 2006, with an emphasis on qualitative comparisons with the model results due to the different time periods and spatial regions considered. The intention is to identify whether the correlations and lag times are broadly consistent to build confidence that the model is capable of capturing the physics of the diurnal evolutions.

The observational correlations and lag times (Table 3) are calculated in an identical manner to the model data, but the second PC of CTH in the SEVIRI observations is used. This is because the first two PCs are reversed when compared to the model PCs, and after examining the corresponding EOF it was clear that this second pattern contained the convective patterns that we are interested in. The observations, just like the model, show that the magnitude of the maximum correlation coefficient with the CTH PC is larger for TOA albedo PC2 than OLR PC2. The lag time between the OLR PC2 and the CTH PC is 0.8 hours longer in the observations than the model. For the TOA albedo, the lag time is the same. This supports the model finding that the TOA albedo responds more rapidly to cloud development than OLR and, if anything, suggests that the difference is even larger than the model indicates.

## 5 Summary and conclusions

The diurnal cycle, a fundamental forcing cycle for our weather and climate, has been assessed using global output of Earth's outgoing energy flows in September 2010 from the Met Office Unified Model. Models have the unique ability to generate spatially complete, high temporal resolution data fields for a wide variety of geophysical variables simultaneously, unrivalled by current observations. Dominant patterns of variability have been extracted from the thermally-emitted and solar-reflected components of the outgoing energy flows, and the extent of coupling of these patterns with the variability in other relevant geophysical variables examined.

The two dominant patterns of diurnal variability in the emitted longwave component are found to be consistent with solar heating of the land surface and development of convective cloud, respectively. The first pattern is highly coupled with variations in the surface temperature and the second pattern is highly coupled with variations in cloud water and height, further supporting the physical attributions. These patterns represent the first fully global estimates of the dominant patterns of diurnal variability in the emitted radiation from our planet, but are similar to those found in previous studies that used spatially-limited satellite observations. The amount of variance explained by the two dominant patterns here is 68.5 % and 16.0 %, respectively, totalling 84.5 %. This is markedly less than that previously found in observations over Africa, Europe and surrounding waters, with around half of the difference resulting from the different spatial regions considered. This demonstrates the importance of complete global coverage if revealing the relative importance of diurnal processes controlling the longwave component of the global energy budget is of interest.

The two dominant patterns of diurnal variability in the reflected shortwave component, calculated in terms of albedo, are found to be consistent with a dependence on the angle of the incoming solar radiation and the development of both convective



and marine stratocumulus cloud, respectively. The dependence due to the angle of the incoming solar radiation explains 88.4 % of the diurnal variance alone, and is found to be a result of contributions from changing surface and cloud albedo, as well as enhanced scattering from aerosols and atmospheric molecules. Atmospheric absorption acts to reverse the enhanced

scattering at larger solar zenith angles, but is around an order of magnitude less influential than the scattering under typical clear-sky conditions. For the second pattern related to cloud development, the spatial variability is very similar to the equivalent longwave pattern in convective regions and is also strongly coupled to the variability in cloud water and height in these regions. However, there are substantial additional sources of compensating variability in marine stratocumulus regions. This demonstrates why both the longwave and shortwave components are required together for revealing all important diurnal

radiative processes. This second pattern only explains 6.7 % of the total variance, suggesting that cloud development and dissipation are relatively less important in controlling the diurnal variability of reflected shortwave radiation. The equivalent patterns from geostationary satellite observations centred over Africa are also presented, which repeat the dominance and features of the first modelled pattern, and the presence of compensating convective and marine cloud variations in the second modelled pattern, indicating that the physical processes dominating the diurnal variability in the modelled reflected solar

radiation are robust.

The strong coupling between radiation and cloud variability is only achieved with significant lag times between the variables. The lag times between convective patterns in emitted longwave radiation and cloud variables paint a coherent picture. Initial development of low-altitude liquid cloud is followed later by development of ice cloud at higher altitudes which is in turn followed by development of high-level anvil cloud. Evaporation of this anvil cloud into the surroundings moistens

the upper troposphere and appears to delay the longwave radiation response to the reduction in cloud height by several hours. For the shortwave pattern related to cloud development the lag times with the same cloud variables are substantially shorter. The moist upper troposphere does not continue to enhance the reflected shortwave radiation once the convective cloud dissipates and the additional influence of marine stratocumulus cloud pulls the pattern closer to the cloud variations. The result is that the shortwave radiation response to diurnal cloud development and dissipation is sharper and more immediate than the

longwave response, which is supported by the equivalent patterns in satellite observations.

Interpreted from a broader perspective, these results demonstrate that a multi-variable, high temporal resolution and complete coverage approach can lead to enhanced process understanding of Earth. This highlights a profound gap and a need towards observing systems capable of observing everything, everywhere, all of the time. The patterns identified in this study could help refine sampling strategies to maximize diurnal information obtained from such observations, and we fully support

the call for global, diurnal observing systems for Earth outgoing radiation in the future.

**Data availability.** The modelled fields used in this study have been archived at the Met Office and are available upon request from the authors. The GERB data (GERB Edition 1 HR product) are available via online download from the Centre for Environmental Data Analysis (CEDA) at http://catalogue.ceda.ac.uk/uuid/d8a5e58e59eb31620082dc4fd10158e2. Here we

have applied the "SW combined adjustment" outlined in the processing document also available from the CEDA. The SEVIRI





CMSAF CLAAS Edition 2 data are available via FTP after registering for the CMSAF Web User Interface. The order page for the CTX product used here can be found at https://wui.cmsaf.eu/safira/action/viewProduktDetails?eid=21235&fid=15. Links checked and working as of 23 Nov 2017.

**Acknowledgments.** This work was supported by the Natural Environment Research Council (NERC) SCience of the Environment: Natural and Anthropogenic pRocesses, Impacts and Opportunities (SCENARIO) Doctoral Training Partnership (DTP), grant number NE/L002566/1. The authors would like to thank Richard Allan, Keith Shine and Graeme Stephens for useful suggestions throughout this study.

**Competing interests.** The authors declare that they have no conflict of interest.

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



**Table 1.** Maximum correlation coefficient and time lag at which it occurs from a cross-correlation between principal components (PCs) of modelled outgoing longwave radiation (OLR) and PCs of modelled surface temperature, cloud liquid water path (LWP), cloud ice water path (IWP) and cloud top height (CTH).

|  | Correlation coefficient | Time lag (hours) |
|---|---|---|
| OLR PC1 vs. surface temperature PC1 | 0.998 | −0.2 |
| OLR PC2 vs. LWP PC1 | −0.983 | 4.6 |
| OLR PC2 vs. IWP PC1 | −0.978 | 3.4 |
| OLR PC2 vs. CTH PC1 | −0.969 | 3.0 |





**Table 2.** Same as Table 1, but for modelled top-of-atmosphere (TOA) albedo.

|  | Correlation coefficient | Time lag (hours) |
|---|---|---|
| TOA albedo PC2 vs. LWP PC1 | 0.997 | 2.8 |
| TOA albedo PC2 vs. IWP PC1 | 0.990 | 1.8 |
| TOA albedo PC2 vs. CTH PC1 | 0.998 | 1.0 |




**Table 3.** Same as Table 1, but for OLR and TOA albedo retrieved from Geostationary Earth Radiation Budget (GERB) observations, and CTH retrieved from Spinning Enhanced Visible and InfraRed Imager (SEVIRI) observations.

|  | Correlation coefficient | Time lag (hours) |
|---|---|---|
| GERB OLR PC2 vs. SEVIRI CTH PC2 | −0.961 | 3.8 |
| GERB TOA albedo PC2 vs. SEVIRI CTH PC2 | 0.992 | 1.0 |






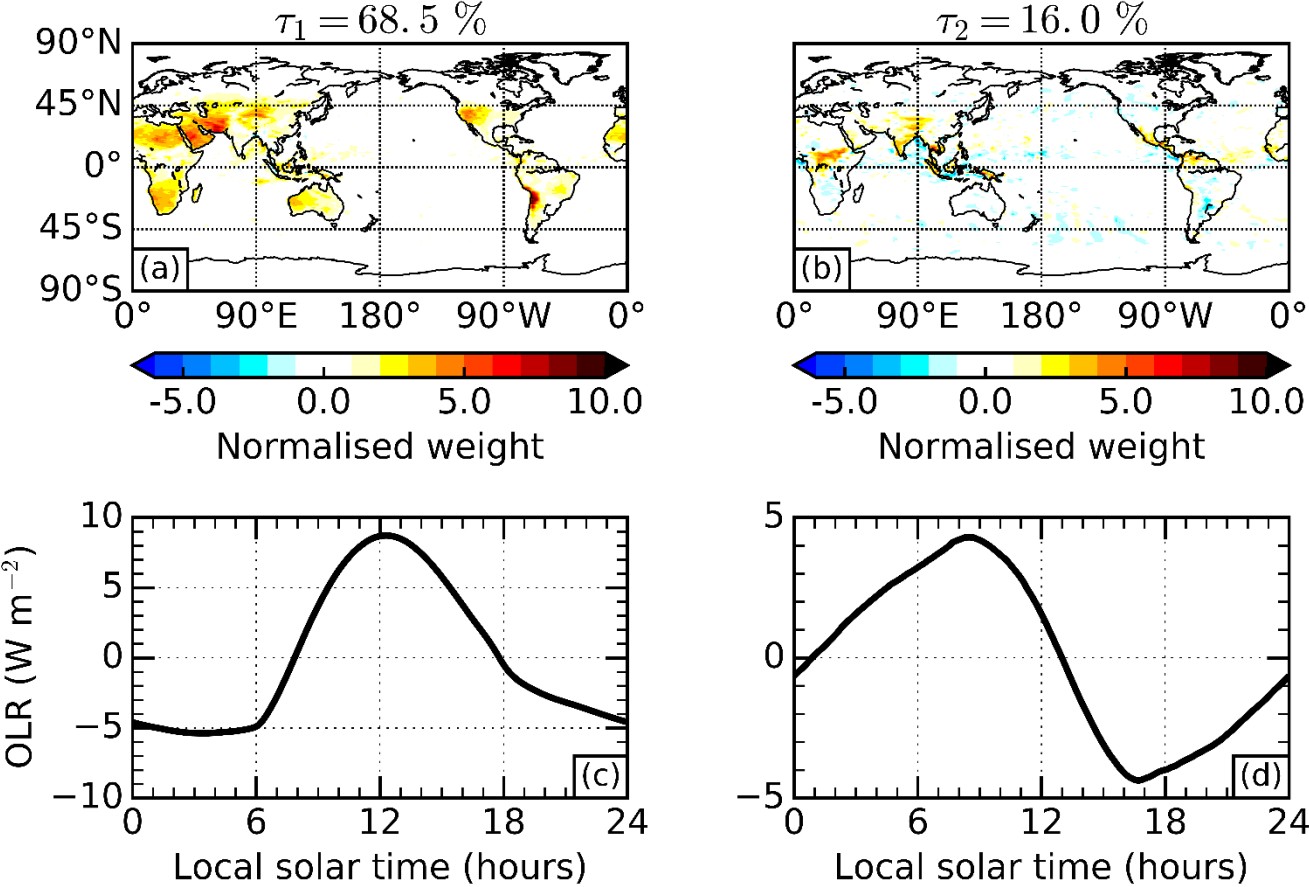

**Figure 1.** Principal component analysis of the global outgoing longwave radiation (OLR) diurnal cycle for September 2010 in the Met Office model. The empirical orthogonal functions (**(a)** and **(b)**) and principal components (**(c)** and **(d)**) are presented for the first (**(a)** and **(c)**) and second (**(b)** and **(d)**) most dominant patterns of variability. The percentage variance explained by each pattern is stated above the corresponding empirical orthogonal function.



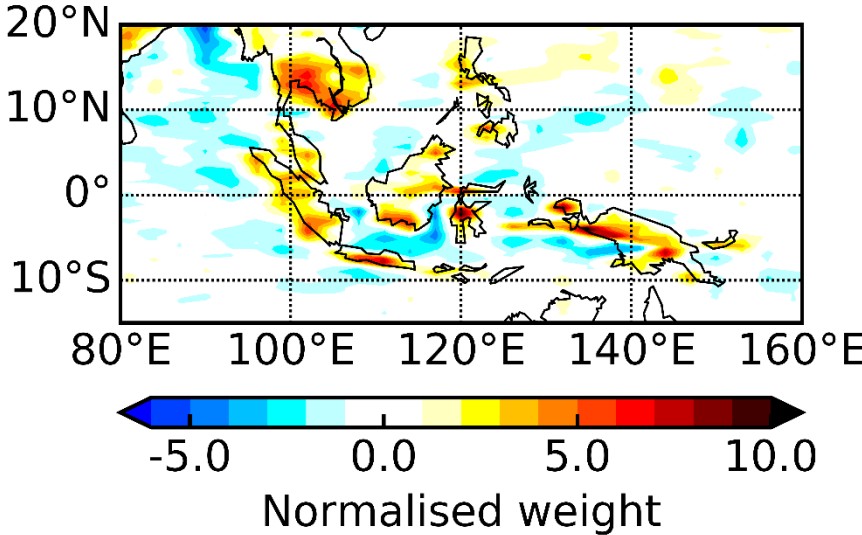

**Figure 2.** A zoom-in of Fig. 1b showing the second empirical orthogonal function over the Maritime Continent region bounded by 15° S–20° N and 80° E–160° E.



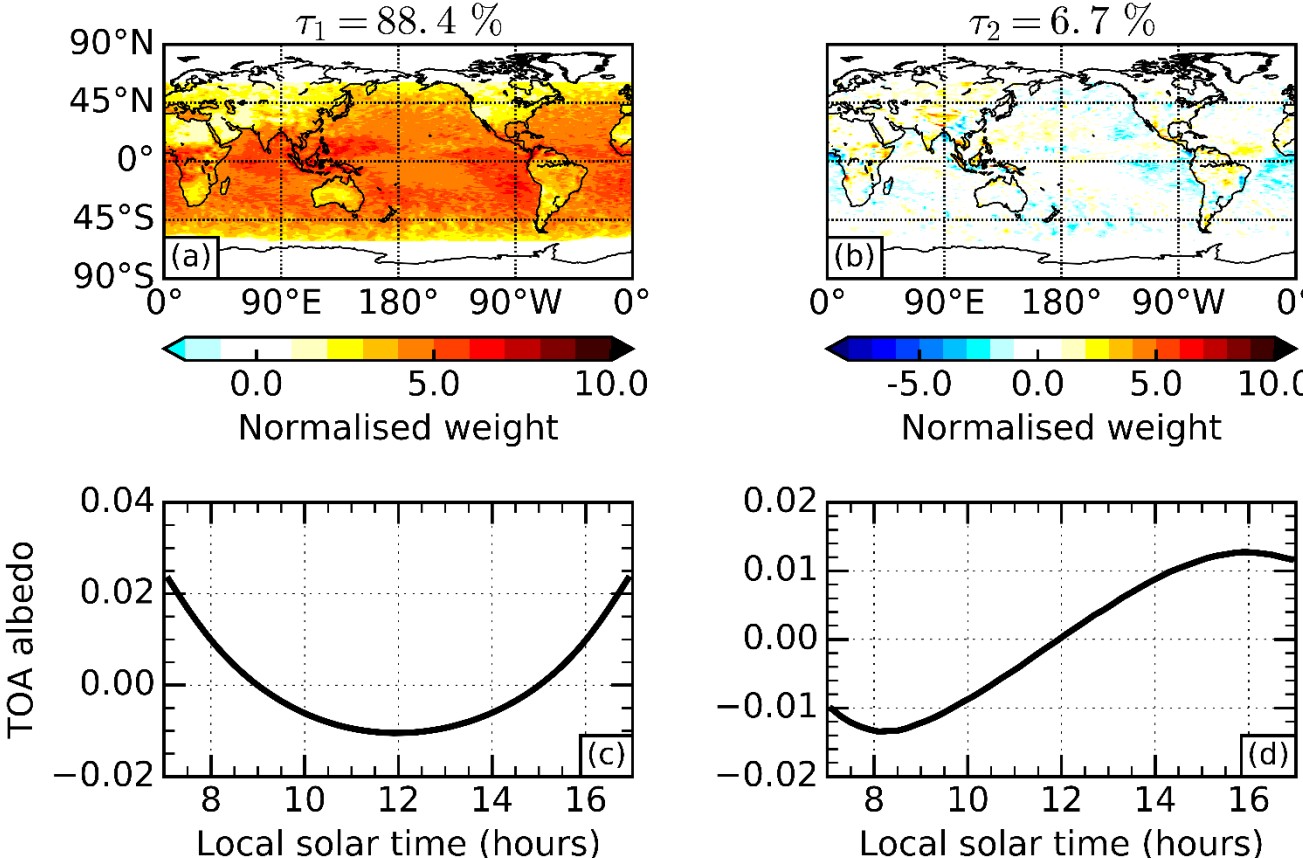

**Figure 3.** Same as Fig. 1, but for the top-of-atmosphere (TOA) albedo.



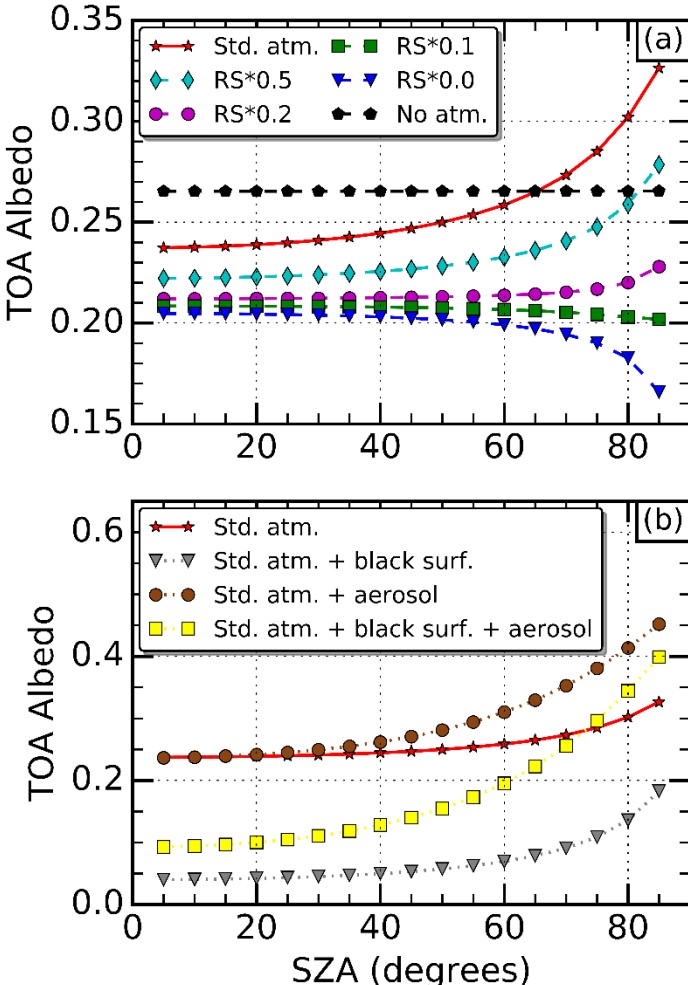

**Figure 4.** Simulation of top-of-atmosphere (TOA) albedo as a function of solar zenith angle (SZA). The dashed lines **(a)** represent atmospheres where a scaling factor has been applied to the Rayleigh scattering (RS). The dotted lines **(b)** represent atmospheres where either aerosols are included, or the surface albedo is set to zero (black surf.), or both. The solid red line with star marker appearing in both plots represents the standard atmosphere (Std. atm.) with no modifications. All simulations assume a US62 standard atmosphere over a Lambertian vegetated surface unless otherwise stated. When aerosols are included, their optical depth is set to 1 at 550 nm and their optical properties are typical of rural aerosols. Details of the aerosol optical properties and the tool used to perform these calculations, the Discrete Ordinate Radiative Transfer (DISORT) model Santa Barbra DISORT Atmospheric Radiative Transfer, are given by Ricchiazzi et al. (1998).



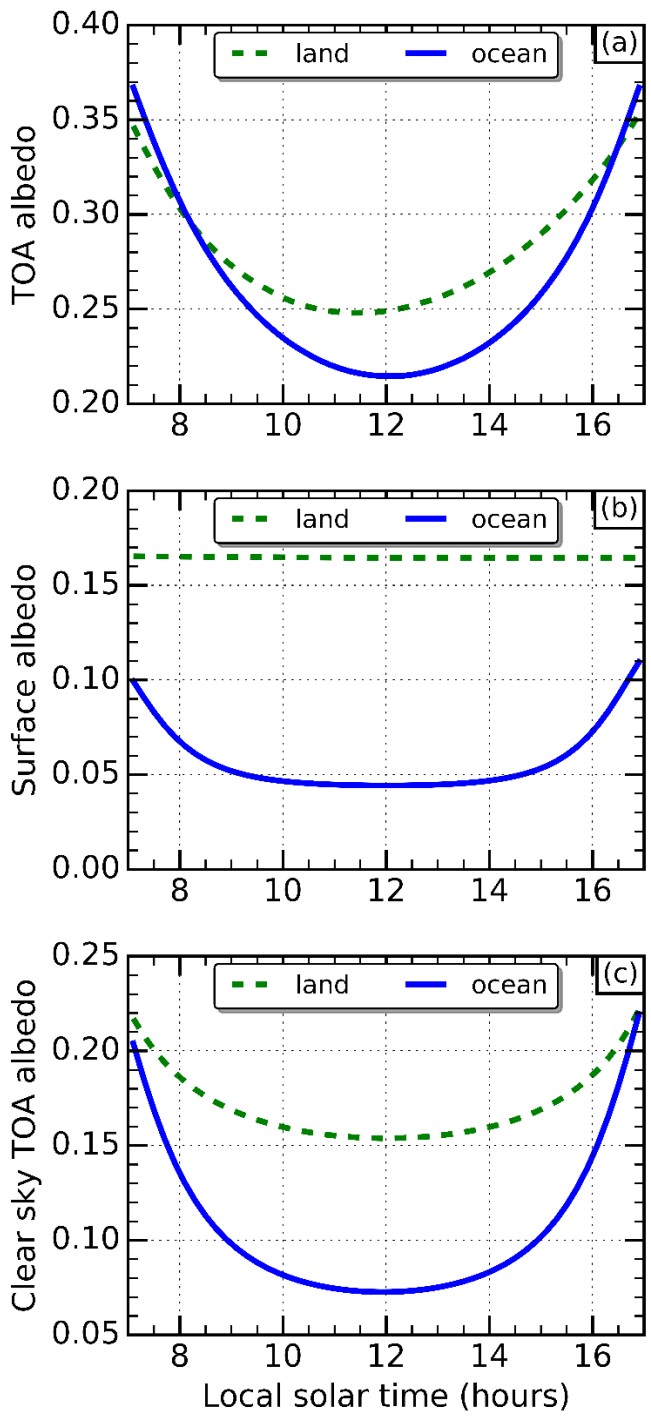

**Figure 5.** The diurnal cycle of global mean albedo for September 2010 in the Met Office model for **(a)** top-of-atmosphere
(TOA) all-sky, **(b)** surface and **(c)** TOA clear-sky, separated over land (green dash) and ocean (blue solid).



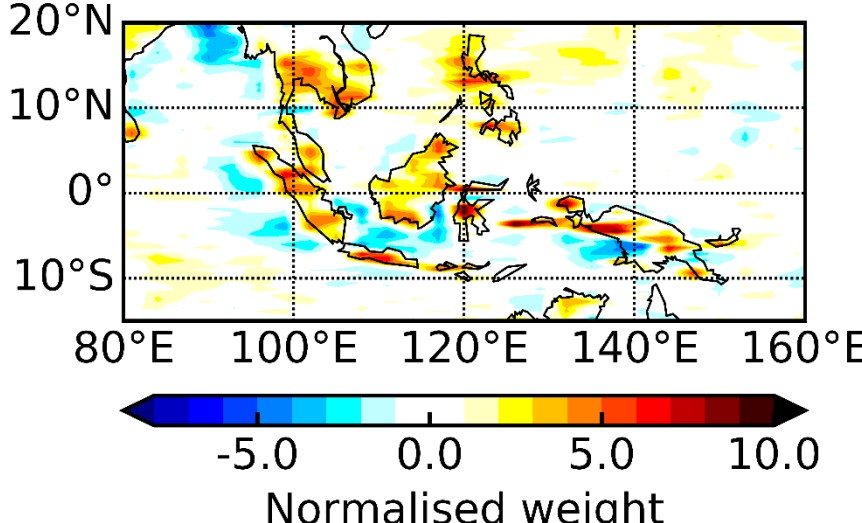

**Figure 6.** A zoom-in of Figure 3b showing the second empirical orthogonal function over the Maritime Continent region bounded by 15° S–20° N and 80° E–160° E.



**Figure 7.** Principal component analysis of the top-of-atmosphere (TOA) albedo diurnal cycle for July 2006 in observations from the Geostationary Earth Radiation Budget instrument. The empirical orthogonal functions ((**a**) and (**b**)) and principal components ((**c**) and (**d**)) are presented for the first ((**a**) and (**c**)) and second ((**b**) and (**d**)) most dominant patterns of variability. The percentage variance explained by each mode is stated above the corresponding empirical orthogonal function.





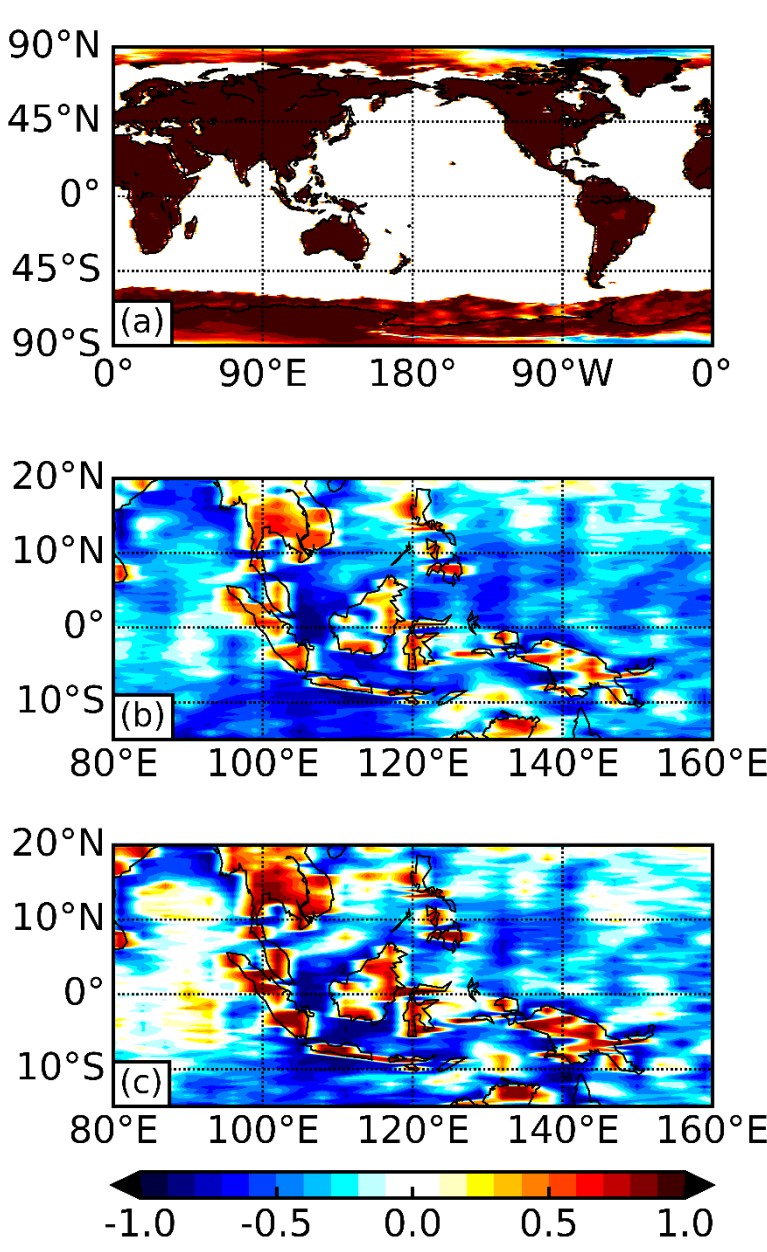

**Figure 8.** Heterogeneous correlation maps for **(a)** the first principal component of outgoing longwave radiation and global surface temperature, **(b)** the second principal component of outgoing longwave radiation and cloud liquid water path in the Maritime Continent region (reversed in sign to aid comparisons) and **(c)** the second principal component of top-of-atmosphere albedo and cloud liquid water path in the Maritime Continent region.



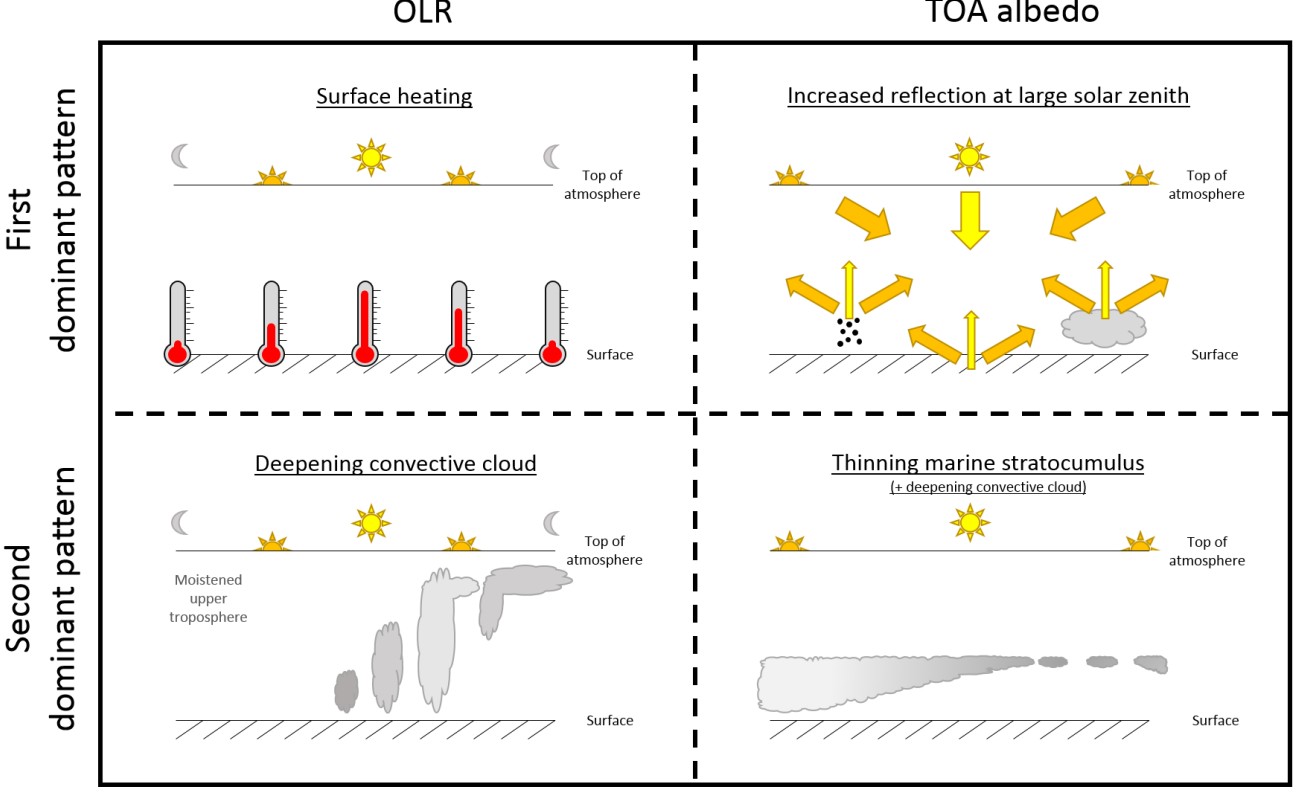

695

**Figure 9.** A schematic diagram showing the processes that control the first **(top)** and second **(bottom)** most dominant patterns in the diurnal variability of the outgoing longwave radiation (OLR) **(left)** and top-of-atmosphere (TOA) albedo **(right)**. Different arrow and Sun colors illustrate the change in solar zenith angle during the diurnal cycle and should not be interpreted as a change in wavelength. The separation of aerosol, surface and cloud reflection in the top right panel is for illustrative 700 purposes only and does not relate to different parts of the diurnal cycle.