# Peer review of "Insights into the diurnal cycle of global Earth outgoing radiation using a numerical weather prediction model"

_Atmospheric Chemistry and Physics, 2017_

## Referee Comment (RC1) · Anonymous Referee #1 · 8 Feb 2018

Manuscript summary:

This study describes the results of an analysis of the diurnal cycle of the Earth outgoing radiation (EOR). A weather prediction model is used as the main tool, but comparisons are also done using satellite data. The diurnal cycle of the EOR and its individual components is analyze using empirical orthogonal functions and principle component analysis. Further the authors tried to correlate the diurnal cycles of EOR with other possibly relevant physical parameters like cloud parameters. The manuscript gives well-described insights into the diurnal cycle of EOR on a global scale.

Review Summary:

[Figure]

The manuscript is well written and presents relevant research on the Earth Outgoing Radiation, that is important for analyzing and understanding the Earth's energy balance. Different data sources are used and the results are well described and discussed. The analysis is only based on 1 month of data, so that the results may partly not represent a climatological behaviour of the diurnal cycle. For example the influence of cloud diurnal cycles may vary from month to month even when globally averaged. This fact is also mentioned in the manuscript, and leads partly to results that should be mainly seen qualitatively, which are still of relevance and interest. In general it should be mentioned even more clearly, that the results may strongly depend on the model used, even though the used Met Office model seems to deliver a reasonable behaviour of the diurnal cycle, which is remarkable as especially the diurnal cycle of clouds is a known weakness in climate and weather models. Overall the manuscript needs only minor revisions.

Review details: (L=line, f=following)

L.39: "lies at the heart of" – please use another formulation!

L.41: "the incoming solar radiation" – better say "the TOA incoming solar radiation", to be more precise

L.46: "discrepancies highlight a lack of understanding" – I think it is not only a lack of understanding that is responsible for the discrepancies between observations and models, it is also a lack of computer power resources to run convective permitting models.

L.46: "yet it is essential we can correctly represent" – sounds wrong –> better say "yet it is essential to correctly represent"

L.71: I would not say "undoubtedly" here. I have seen models that totally missed the observed diurnal cycle of clouds, which meant that no understanding at the process level was possible using this model.

[Figure]

L.132: When mentioning the CLAAS-2 data record, please cite also: - Finkensieper, Stephan; Meirink, Jan-Fokke; van Zadelhoff, Gerd-Jan; Hanschmann, Timo; Benas, Nikolaos; Stengel, Martin; Fuchs, Petra; Hollmann, Rainer; Werscheck, Martin (2016): CLAAS-2: CM SAF CLoud property dAtAset using SEVIRI - Edition 2, Satellite Application Facility on Climate Monitoring, DOI:10.5676/EUM_SAF_CM/CLAAS/V002, https://doi.org/10.5676/EUM_SAF_CM/CLAAS/V002.

L.240 to L.243: According to Fig 1d, does this mean that the diurnal cycle of clouds over land dominates over the diurnal cycle of clouds over ocean ?

L.258: "for a select few regions" sounds wrong.

L.287f: "As a result, the first EOF (Fig 3a) exhibits positive weights in many different predominantly cloud-free regions, such as the global deserts"; Either I did not get the point or something is wrong here. According to Fig 3a, the cloud-free regions, like the Sahara desert, exhibit only very small positive weights, if positive at all.

L.317: "which appears to be captured by the model." – this is a process that is relatively well represented in weather and climate models, which is in line with findings of Pfeifroth et. al, 2012, whom you might cite at this point (https://dx.doi.org/10.1127/0941-2948/2012/0423).

L.372: "is consistent with the lifecycle of a convective system"; Please be aware that this may be a too simplified description. Different types of convective systems exist in the troposphere. Some are locally initiated; and these are the ones that are referred to in this study. However, there are for example also mesoscale convective systems (MCS), which my have a totally different life cycle, and may live for multiple days.

L.421: "because the first two PCs are reversed when compared". How does this come? This is a bit confusing, and if it is only for a technical reason, this fact might be left out completely.

L.472: "understanding of Earth." – something seems to missing here.

---

## Referee Comment (RC2) · Anonymous Referee #2 · 9 Feb 2018

General Comments:

In this paper the authors investigate the diurnal cycle of Earth's outgoing radiation(EOR), splitting its components into outgoing longwave radiation (OLR) and reflected shortwave radiation (RSR). Their primary focus is on analyzing the output from the Met Office NWP model for the month of September 2010 and GEBA output for July 2006 using Principal component analysis (PCA). For each EOR component they investigate the cause of the first two EOFs. In the case of OLR they claim that the first EOF, which is the dominant signal, is largely related to changes in surface/atmospheric temperature, while the second is related to the diurnal cycle of deep convection. In the

case of RSR, the first EOF is again dominant and is controlled by the atmospheric path length, while the second is related to the timing of deep and shallow convection.

I found the paper to be well written and the analysis clearly presented. I think that the authors have achieved their aim of showing the dominant signals that influence the diurnal cycle of EOR. It is also interesting to see the reasonably good agreement between the NWP and observations. To this end I have no issue with recommending the paper for publication following minor revisions. I do think though the paper would benefit from a more detailed analysis of the surface versus atmospheric contribution to the first OLR EOF. It feels like the detailed analysis that went into understanding the radiative transfer leading to the RSR signal has not been replicated in the case of OLR. I detail my concerns below.

Specific comments:

Lines 88: I think a few more sentences discussing the impact of fixed sea surface temperatures is needed here. I know it is discussed later on, but the fact there is no diurnal SST cycle is quite a major caveat.

Line 139: I understand that it may not be possible to analyse the satellite data at the equinox, but it would seem that it would at least be possible to analyse the the NWP output for the same month as the satellite. This would lead to a cleaner comparison. If this is not possible, then perhaps explain in more detail why this is the case.

In general, one weaknesses of the paper is the fact the authors only look at one month of one year. Hence the need for more clarity about why just one month is looked at and some text expressing the limitations this imposes would be useful. What would the authors expect different in their results if they did same analysis with 30 years of monthly data?

Lines 230:236. I think that the authors have to dig a bit deeper here. It should be relatively straightforward to use a RT code to distinguish how much of this OLR signal

is due to the surface compared to the atmosphere. This will help improve our understanding of whether the surface, boundary layer or lower/mid troposphere diurnal cycle of temperature is most important for understanding the diurnal cycle of OLR. The 10% number from Costa and Shine used here may also be misleading, as locally these numbers can be bigger and I suspect are bigger over the dry land regions that have the strongest diurnal signal shown in figure 1. Furthermore, even if the surface only accounts for 10% of the absolute OLR signal, a 25 K swing in surface temperature could still cause a big swing in OLR. Consider a change from 300 K to 325 K = 0.1*5.67E-8(325^4 -300^4) =17.3 Wm-2. Another issue with the claim that atmospheric temperature is important is that most of the emission from the lower atmosphere to space is dominated by emission from the H2O self continuum. However, the optical depth of the continuum scales with the square of vapour pressure and may be quite weak in dry hot regions. This again makes me think that the atmospheric contribution from the dominate regions highlighted in fig 1 might be smaller than that of the surface. Hence, I am not convinced by the term 'large fraction' used in the statement on line 235. Given that the paper aims to provide insight into the mechanism behind the EOR signal means that this 'fraction' should be quantified. I therefore encourage the authors to perform a few simple RT runs, even using idealized atmospheres, so to make the attribution of the OLR signal clearer.

.

Line 281: I generally like that the analysis (i.e. Fig 4) that the authors have performed on investigating the causes behind the 'U' shape. However, the impact of aerosol and mean cloudiness could be dealt with a bit better. Here the authors say they use only one aerosol case; 'rural aerosol' to see how aerosol loading could change the relationship between SZA and TOA albedo. Would it not be more useful to look at the extremes between say a highly scattering aerosol environment(e.g. high SO4 or sea salt) versus a highly absorbing aerosol environment (e.g black carbon). Just using one simple aerosol case does not really provide much insight into how much aerosol

can alter the diurnal cycle of RSR. Also I wonder about impact of the mean state of cloudiness (as opposed to the diurnal cycle). I suspect that this 'U' shape would be stronger for cloudy versus clear regions (as shown in fig 5), but may get weaker as the mean cloudiness of a region goes up. That is because the amount of radiation scattered to space per unit optical depth decreases with increasing cloud optical depth.

Line 335: I would place more emphasis on this result in the abstract and conclusions. The fact that your technique of analyzing the diurnal cycle highlights some clear limitations of the NWP cloud fields is an important result.

---

## Author Response (AR1)

Dear editor and reviewers,

Thank you for taking the time to review our manuscript titled "Insights into the diurnal cycle of global Earth outgoing radiation using a numerical weather prediction model" (acp-2017-1144). We are pleased to hear your recommendations for minor revisions and appreciative of the thoughtful comments and suggestions. Please find our responses below which address your comments point by point and list the corresponding revisions made to the manuscript. Any line numbers stated in the author responses correspond to those in the revised manuscript.

Yours faithfully,

Jake Gristey (on behalf of all authors)

**Response to Anonymous Referee #1**

| Overall assessment                                                                                                                                                                                                                                                                                                                                                                                                                                                                                                                                                                                                                                                                                                                                                                                                                                                                                                                                                                                                                                                                                     | Response                                                                                                                                                                                                                                                                                                                                                                                                                                                                                                                                                                                                                                                                                                                                                                                                                                                                                                                                                                                                                                                                                                                                                   |  |  |  |  |
|--------------------------------------------------------------------------------------------------------------------------------------------------------------------------------------------------------------------------------------------------------------------------------------------------------------------------------------------------------------------------------------------------------------------------------------------------------------------------------------------------------------------------------------------------------------------------------------------------------------------------------------------------------------------------------------------------------------------------------------------------------------------------------------------------------------------------------------------------------------------------------------------------------------------------------------------------------------------------------------------------------------------------------------------------------------------------------------------------------|------------------------------------------------------------------------------------------------------------------------------------------------------------------------------------------------------------------------------------------------------------------------------------------------------------------------------------------------------------------------------------------------------------------------------------------------------------------------------------------------------------------------------------------------------------------------------------------------------------------------------------------------------------------------------------------------------------------------------------------------------------------------------------------------------------------------------------------------------------------------------------------------------------------------------------------------------------------------------------------------------------------------------------------------------------------------------------------------------------------------------------------------------------|--|--|--|--|
|                                                                                                                                                                                                                                                                                                                                                                                                                                                                                                                                                                                                                                                                                                                                                                                                                                                                                                                                                                                                                                                                                                        | • Thank you very much for your positive review.                                                                                                                                                                                                                                                                                                                                                                                                                                                                                                                                                                                                                                                                                                                                                                                                                                                                                                                                                                                                                                                                                                            |  |  |  |  |
| Manuscript summary:
This study describes the results of an analysis of the
diurnal cycle of the Earth outgoing radiation (EOR).
A weather prediction model is used as the main
tool, but comparisons are also done using satellite
data. The diurnal cycle of the EOR and its individual
components is analyze using empirical orthogonal
functions and principle component analysis. Further
the authors tried to correlate the diurnal cycles of
EOR with other possibly relevant physical
parameters like cloud parameters. The manuscript
gives well-described insights into the diurnal cycle of
EOR on a global scale.                                                                                                                                                                                                                                                                                                                                                                                                                                       | • Thank you for the summary, which nicely listed the key messages of the manuscript.                                                                                                                                                                                                                                                                                                                                                                                                                                                                                                                                                                                                                                                                                                                                                                                                                                                                                                                                                                                                                                                                       |  |  |  |  |
| Review Summary:
The manuscript is well written and presents relevant
research on the Earth Outgoing Radiation, that is
important for analyzing and understanding the
Earth's energy balance. Different data sources are
used and the results are well described and
discussed. The analysis is only based on 1 month of
data, so that the results may partly not represent a
climatological behaviour of the diurnal cycle. For
example the influence of cloud diurnal cycles may
vary from month to month even when globally
averaged. This fact is also mentioned in the
manuscript, and leads partly to results that should
be mainly seen qualitatively, which are still of
relevance and interest. In general it should be
mentioned even more clearly, that the results may
strongly depend on the model used, even though
the used Met Office model seems to deliver a
reasonable behaviour of the diurnal cycle of clouds
is a known weakness in climate and weather
models. Overall the manuscript needs only minor
revisions. |  <li>Thank you for your positive comments on the quality of the writing and discussion.</li> <li>We agree that the results should be interpreted qualitatively when considering the climatological behaviour since they are only based on one month of data. You are also correct that the results should still be of relevance. This is because the seasonal variations in the second principle component, representing the cloud diurnal cycles that you mention, are typically much smaller than the total signal. See, for example, Fig. 13 from Rutan et al. (2014) below that demonstrates the relatively small seasonal variations, albeit over land only and not global. "Normalized Day" in this figure is -1.0 at sunrise, 1.0 at sunset, and 0.0 at solar noon.</li> <li>For 13. Second principal component for each season over land only and not surface only.</li> <li>Fto 13. Second principal component for each season over land on the model used. We have added a comment in the conclusions on L462-463 "While the characteristics of the diurnal cycle will depend on the model chosen" to make this explicitly clear.</li>  |  |  |  |  |

**Response to Anonymous Referee #1 (cont.)**

| Specific comments                                                                                                                                                                                                                                                                                                                                                                                                                                                                        | Response                                                                                                                                                                                                                                                                                                                                                                                                                                          |  |  |  |  |
|------------------------------------------------------------------------------------------------------------------------------------------------------------------------------------------------------------------------------------------------------------------------------------------------------------------------------------------------------------------------------------------------------------------------------------------------------------------------------------------|---------------------------------------------------------------------------------------------------------------------------------------------------------------------------------------------------------------------------------------------------------------------------------------------------------------------------------------------------------------------------------------------------------------------------------------------------|--|--|--|--|
| L.39: "lies at the heart of" – please use another formulation!                                                                                                                                                                                                                                                                                                                                                                                                                           | • OK. Changed from "lies at the heart of" to "underpins" on L39.                                                                                                                                                                                                                                                                                                                                                                                  |  |  |  |  |
| L.41: "the incoming solar radiation" – better say "the TOA incoming solar radiation", to be more precise                                                                                                                                                                                                                                                                                                                                                                                 | • OK. Added "TOA" on L41.                                                                                                                                                                                                                                                                                                                                                                                                                         |  |  |  |  |
| L.46: "discrepancies highlight a lack of
understanding" – I think it is not only a lack of
understanding that is responsible for the
discrepancies between observations and models, it
is also a lack of computer power resources to run
convective permitting models.                                                                                                                                                                                                    | • Again, this is a fair point and the authors agree.
Added "along with insufficient computing
resources" on L48.                                                                                                                                                                                                                                                                                                                            |  |  |  |  |
| L.46: "yet it is essential we can correctly represent"
– sounds wrong –> better say "yet it is essential to
correctly represent"                                                                                                                                                                                                                                                                                                                                                   | • OK. Changed from "we can" to "to" on L48.                                                                                                                                                                                                                                                                                                                                                                                                       |  |  |  |  |
| L.71: I would not say "undoubtedly" here. I have
seen models that totally missed the observed
diurnal cycle of clouds, which meant that no
understanding at the process level was possible
using this model.                                                                                                                                                                                                                                                                 | • OK. Removed "undoubtedly".                                                                                                                                                                                                                                                                                                                                                                                                                      |  |  |  |  |
| L.132: When mentioning the CLAAS-2 data record,
please cite also: - Finkensieper, Stephan; Meirink,
Jan-Fokke; van Zadelhoff, Gerd-Jan; Hanschmann,
Timo; Benas, Nikolaos; Stengel, Martin; Fuchs, Petra;
Hollmann, Rainer; Werscheck, Martin (2016):
CLAAS-2: CM SAF CLoud property dAtAset using
SEVIRI - Edition 2, Satellite Application Facility on
Climate Monitoring,
DOI:10.5676/EUM_SAF_CM/CLAAS/V002,
https://doi.org/10.5676/EUM_SAF_CM/CLAAS/V002 | • Thank you for the suggested reference. Added
"Finkensieper et al., 2016" on L132 and added
"Finkensieper, S., Meirink, JF., van Zadelhoff,
GJ., Hanschmann, T., Benas, N., Stengel, M.,
Fuchs, P., Hollmann, R., Werscheck, M.: CLAAS-
2: CM SAF CLoud property dAtAset using
SEVIRI - Edition 2, Satellite Application Facility
on Climate Monitoring,
doi:10.5676/EUM_SAF_CM/CLAAS/V002,
2016." to reference list. |  |  |  |  |
| L.240 to L.243: According to Fig 1d, does this mean
that the diurnal cycle of clouds over land dominates
over the diurnal cycle of clouds over ocean ?                                                                                                                                                                                                                                                                                                                             | • This is true, but Fig. 1d does not show this, Fig 1b does. The sign of Fig. 1b and 1d could both be flipped and it would be equally valid. It is the larger magnitude of EOF weights over land than ocean in Fig. 1b that indicates a stronger signal over land (clearer in Fig. 2).                                                                                                                                                            |  |  |  |  |
| L.258: "for a select few regions" sounds wrong.                                                                                                                                                                                                                                                                                                                                                                                                                                          | • OK. Changed "select few" to "small number of" on L289.                                                                                                                                                                                                                                                                                                                                                                                          |  |  |  |  |

| L.287f: "As a result, the first EOF (Fig 3a) exhibits
positive weights in many different predominantly
cloud-free regions, such as the global deserts";
Either I did not get the point or something is wrong
here. According to Fig 3a, the cloud-free regions, like
the Sahara desert, exhibit only very small positive                                                                                                       | • This sentence has been updated for clarity. The emphasis should be on the fact that the signal is weakly positive (ie. yellow-ish colours) in these regions. We have checked the values and they are rarely negative anywhere in this EOF. Added "weakly" on L319.                                                                                                                                                                     |  |  |  |
|-----------------------------------------------------------------------------------------------------------------------------------------------------------------------------------------------------------------------------------------------------------------------------------------------------------------------------------------------------------------------------------------------------------------------------------------------|------------------------------------------------------------------------------------------------------------------------------------------------------------------------------------------------------------------------------------------------------------------------------------------------------------------------------------------------------------------------------------------------------------------------------------------|--|--|--|
| weights, if positive at all.                                                                                                                                                                                                                                                                                                                                                                                                                  |                                                                                                                                                                                                                                                                                                                                                                                                                                          |  |  |  |
| L.317: "which appears to be captured by the
model." – this is a process that is relatively well
represented in weather and climate models, which
is in line with findings of Pfeifroth et. al, 2012, whom
you might cite at this point
(https://dx.doi.org/10.1127/0941-2948/2012/0423).                                                                                                                                       |  <li>Thank you for the suggested reference. Added "a process that is relatively well represented in weather and climate models (Pfeifroth et al., 2012) and whichhere" on L350-351 and added "Pfeifroth, U., Hollmann, R., and Ahrens, B.: Cloud cover diurnal cycles in satellite data and regional climate model simulations, Meteorol. Z., 21(6), 551–560, doi: 10.1127/0941-2948/2012/0423, 2012." to reference list.</li>  |  |  |  |
| L.372: "is consistent with the lifecycle of a convective system"; Please be aware that this may be a too simplified description. Different types of convective systems exist in the troposphere. Some are locally initiated; and these are the ones that are referred to in this study. However, there are for example also mesoscale convective systems (MCS), which my have a totally different life cycle, and may live for multiple days. | • Our description is intended to represent the locally initiated, repeating, and diurnally driven convection. Recall that the data considered is the average diurnal cycle for the entire month, supressing transient types of phenomena like MCS. To make this clear, we have added "locally driven" on L407.                                                                                                                           |  |  |  |
| L.421: "because the first two PCs are reversed when
compared". How does this come? This is a bit
confusing, and if it is only for a technical reason, this
fact might be left out completely.                                                                                                                                                                                                                                        | • Yes, this can be considered a technical reason. It happens because the change in percentage variance explained by the patterns between global model data and regional observations changes the order that the PC's appear. To avoid confusion, we have removed the first two sentences of this paragraph. For clarity, we also have updated the method section by removing "leading" and adding "related" on L217.                     |  |  |  |
| L.472: "understanding of Earth." – something seems to missing here.                                                                                                                                                                                                                                                                                                                                                                           | • OK. Changed "enhanced process understanding of Earth" to "an enhanced understanding of processes in the Earth system" on L505                                                                                                                                                                                                                                                                                                          |  |  |  |

| Overall assessment                                                                                                                                                                                                                                                                                                                                                                                                                                                                                                                                                                                                                                                                                                                                                                                                                                                                                                                                                                                                                                                                                                                                                                                                                                                                                                                                                                                                                                                                                                                                                                                                                                                                                                                     | Response                                                                                                                                                                                                                                                                                                                                                                                                                                                                                                  |  |  |  |  |  |
|----------------------------------------------------------------------------------------------------------------------------------------------------------------------------------------------------------------------------------------------------------------------------------------------------------------------------------------------------------------------------------------------------------------------------------------------------------------------------------------------------------------------------------------------------------------------------------------------------------------------------------------------------------------------------------------------------------------------------------------------------------------------------------------------------------------------------------------------------------------------------------------------------------------------------------------------------------------------------------------------------------------------------------------------------------------------------------------------------------------------------------------------------------------------------------------------------------------------------------------------------------------------------------------------------------------------------------------------------------------------------------------------------------------------------------------------------------------------------------------------------------------------------------------------------------------------------------------------------------------------------------------------------------------------------------------------------------------------------------------|-----------------------------------------------------------------------------------------------------------------------------------------------------------------------------------------------------------------------------------------------------------------------------------------------------------------------------------------------------------------------------------------------------------------------------------------------------------------------------------------------------------|--|--|--|--|--|
|                                                                                                                                                                                                                                                                                                                                                                                                                                                                                                                                                                                                                                                                                                                                                                                                                                                                                                                                                                                                                                                                                                                                                                                                                                                                                                                                                                                                                                                                                                                                                                                                                                                                                                                                        | • Thank you very much for your positive review.                                                                                                                                                                                                                                                                                                                                                                                                                                                           |  |  |  |  |  |
| General Comments:
In this paper the authors investigate the diurnal
cycle of Earth's outgoing radiation(EOR), splitting its
components into outgoing longwave radiation (OLR)
and reflected shortwave radiation (RSR). Their
primary focus is on analyzing the output from the
Met Office NWP model for the month of September
2010 and GEBA output for July 2006 using Principal
component analysis (PCA). For each EOR component
they investigate the cause of the first two EOFs. In
the case of OLR they claim that the first EOF, which
is the dominant signal, is largely related to changes
in surface/atmospheric temperature, while the
second is related to the diurnal cycle of deep
convection. In the case of RSR, the first EOF is again
dominant and is controlled by the atmospheric path
length, while the second is related to the timing of
deep and shallow convection.
I found the paper to be well written and the analysis
clearly presented. I think that the authors have
achieved their aim of showing the dominant signals
that influence the diurnal cycle of EOR. It is also
interesting to see the reasonably good agreement
between the NWP and observations. To this end I
have no issue with recommending the paper for
publication following minor revisions. I do think
though the paper would benefit from a more
detailed analysis of the surface versus atmospheric
contribution to the first OLR EOF. It feels like the
detailed analysis that went into understanding the
radiative transfer leading to the RSR signal has not
been replicated in the case of OLR. I detail my
concerns below. |  <li>Thank you for the summary, which nicely listed the key messages of the manuscript</li> <li>Thank you for noting that the manuscript is clear, well written and you feel that we have achieved our aim.</li> <li>Upon reflection, we agree that investigating the contribution from the surface and atmosphere to the first OLR EOF would be an insightful and useful addition, and have now included some additional experiments, as detailed in the specific comments section below.</li>  |  |  |  |  |  |

**Response to Anonymous Referee #2**

| Specific comments                                                                                                                                                                                                                                                                                                                                                                                                                                                                                                                                                                                                                                                                                                                                              | Response • OK. We have added "The sea-surface temperatures are updated daily and, therefore, do not exhibit diurnal variability." on L87-88 to identify this caveat as soon as the dataset is introduced. As you already mention, we have already been careful to state where we believe this could influence our results on L292 and L399-403. • Note that for the GERB OLR PC1 presented in Fig. 1b in Comer et al., 2007 (below), the weight over ocean is very close to zero. In other words, because the diurnal cycle in ocean surface temperature is substantially smaller than that over land, the corresponding OLR signal is completely dominated by land surface temperature variations, even when SST variations are present. Therefore, the lack of diurnal SST cycle in model simulations does not represent a severe issue, at least in terms of the direct emission, as already mentioned on L402-403. • Fgr-1, 6, of First morphrepart composers of CHEB data for the your surface the represent emperature with d, the treemet emperature is substantially smaller than that over land, the corresponding OLR signal is completely dominated by land surface temperature variations, even when SST variations are present. Therefore, the lack of diurnal SST cycle in model simulations does not represent a severe issue, at least in terms of the direct emission, as already mentioned on L402-403. • Fgr-1, 6, of First morphrepart composers of CHEB data for the your when the there expected emperature is model data for the same month as the satellite (July 2006). However, the subtle, but important point of why we have not done this in our study is that away from the equinox, a global analysis is not possible. Since the focus of our study is |  |  |  |  |
|----------------------------------------------------------------------------------------------------------------------------------------------------------------------------------------------------------------------------------------------------------------------------------------------------------------------------------------------------------------------------------------------------------------------------------------------------------------------------------------------------------------------------------------------------------------------------------------------------------------------------------------------------------------------------------------------------------------------------------------------------------------|---------------------------------------------------------------------------------------------------------------------------------------------------------------------------------------------------------------------------------------------------------------------------------------------------------------------------------------------------------------------------------------------------------------------------------------------------------------------------------------------------------------------------------------------------------------------------------------------------------------------------------------------------------------------------------------------------------------------------------------------------------------------------------------------------------------------------------------------------------------------------------------------------------------------------------------------------------------------------------------------------------------------------------------------------------------------------------------------------------------------------------------------------------------------------------------------------------------------------------------------------------------------------------------------------------------------------------------------------------------------------------------------------------------------------------------------------------------------------------------------------------------------------------------------------------------------------------------------------------------------------------------------------------------------------------------------------------------------------------------------------------------------------------------|--|--|--|--|
| Lines 88: I think a few more sentences discussing the
impact of fixed sea surface temperatures is needed
here. I know it is discussed later on, but the fact
there is no diurnal SST cycle is quite a major caveat.                                                                                                                                                                                                                                                                                                                                                                                                                                                                                                                                   |                                                                                                                                                                                                                                                                                                                                                                                                                                                                                                                                                                                                                                                                                                                                                                                                                                                                                                                                                                                                                                                                                                                                                                                                                                                                                                                                                                                                                                                                                                                                                                                                                                                                                                                                                                                       |  |  |  |  |
| Line 139: I understand that it may not be possible to
analyse the satellite data at the equinox, but it
would seem that it would at least be possible to
analyse the the NWP output for the same month as
the satellite. This would lead to a cleaner
comparison. If this is not possible, then perhaps
explain in more detail why this is the case. In
general, one weaknesses of the paper is the fact the
authors only look at one month of one year. Hence
the need for more clarity about why just one month
is looked at and some text expressing the limitations
this imposes would be useful. What would the
authors expect different in their results if they did
same analysis with 30 years of monthly data? | • You are correct that it would be possible to analyse
the model data for the same month as the satellite
(July 2006). However, the subtle, but important
point of why we have not done this in our study is
that away from the equinox, a global analysis is not
possible. Since the focus of our study is
specifically on the global scale, we maintain the
focus of our results on the model data from
September 2010. To explain this, we have added
"We acknowledge that it would be ideal to use
model output from July 2006 to compare with
these observations. However, to fully capitalise on
understanding the diurnal cycle at a global scale, it
is crucial to use the model output for September,
because the relative importance of processes
inferred from a global and a regional scale can be                                                                                                                                                                                                                                                                                                                                                                                                                                                                                                                                                                                                                                                                                                                                                                                                                                                                                                                           |  |  |  |  |

**Response to Anonymous Referee #2 (cont.)**

|                                                                                                                                                                                                                                                                                                                                                                                                                                                                                                                                                                                                                                                                                                |  <li>quite different (as discussed in Sect. 4.1)." on L141-144.</li> <li>Perhaps a follow up study could focus on model validation over limited regions during different months, but such a study would have quite different motivation and aims.</li> <li>The key results from the GERB and model comparisons are:  <li>(1) Leading OLR spatial and temporal patterns are similar</li> <li>(2) Variance explained by first two global model OLR patterns is lower than GERB. When subsampling model data over GERB FOV, the total</li>  </li>                                                                                                                                                                                                                                                                                                                                     |
|------------------------------------------------------------------------------------------------------------------------------------------------------------------------------------------------------------------------------------------------------------------------------------------------------------------------------------------------------------------------------------------------------------------------------------------------------------------------------------------------------------------------------------------------------------------------------------------------------------------------------------------------------------------------------------------------|------------------------------------------------------------------------------------------------------------------------------------------------------------------------------------------------------------------------------------------------------------------------------------------------------------------------------------------------------------------------------------------------------------------------------------------------------------------------------------------------------------------------------------------------------------------------------------------------------------------------------------------------------------------------------------------------------------------------------------------------------------------------------------------------------------------------------------------------------------------------------------------------------|
|                                                                                                                                                                                                                                                                                                                                                                                                                                                                                                                                                                                                                                                                                                |  <li>half way between global model data and GERB observations.</li> <li>(3) Leading TOA albedo spatial and temporal patterns are similar, but movement of ITCZ between months is apparent.</li> <li>(4) Convective and marine stratocumulus albedo patterns show up with slightly later timing in observational patterns.</li> <li>(5) The time lag between GERB OLR/albedo and SEVIRI CTH is consistent with the time lags from model data, and supports the more rapid response of shortwave radiation to cloud variations.</li> <li>Overall, (1)-(5) generally show that the comparisons between July GERB and September model dominant patterns are already satisfactory (at the level we are interested in). If we were to repeat the experiment with model output and observations from the same month, we expect the presented results to at least hold, and probably improve.</li>  |
|                                                                                                                                                                                                                                                                                                                                                                                                                                                                                                                                                                                                                                                                                                | • We agree that care must be taken when interpreting
these results in a climatological sense. In this case
we would encourage only qualitative interpretation,
but the analysis should still be relevant given the
relatively small seasonal variations in the PCs (see
Fig. 13 from Rutan et al. (2014) on page 2 of this
document).                                                                                                                                                                                                                                                                                                                                                                                                                                                                                                                                              |
| Lines 230:236. I think that the authors have to dig a
bit deeper here. It should be relatively
straightforward to use a RT code to distinguish how
much of this OLR signal is due to the surface
compared to the atmosphere. This will help improve
our understanding of whether the surface, boundary
layer or lower/mid troposphere diurnal cycle of
temperature is most important for understanding
the diurnal cycle of OLR. The 10% number from
Costa and Shine used here may also be misleading,
as locally these numbers can be bigger and I suspect
are bigger over the dry land regions that have the
strongest diurnal signal shown in figure 1. |  <li>Thank you for this good point.</li> <li>You are correct that in the predominantly clear sky and dry desert regions, where the EOF weights are largest, the directly transmitted radiation from the surface will be higher than the 10% value stated in the Costa and Shine study. This can actually be seen explicitly in Fig. 2 of the Costa and Shine study. We intended to use this 10% value only to give a general background, but agree that it could be misleading.</li> <li>To dig deeper, we have performed some additional radiative transfer simulations to calculate the fraction of increased surface emission that reaches</li>                                                                                                                                                                                                                                          |

Furthermore, even if the surface only accounts for 10% of the absolute OLR signal, a 25 K swing in surface temperature could still cause a big swing in OLR. Consider a change from 300 K to 325 K = 0.1\*5.67E-8(3254 -3004) =17.3 Wm-2. Another issue with the claim that atmospheric temperature is important is that most of the emission from the lower atmosphere to space is dominated by emission from the H2O self continuum. However, the optical depth of the continuum scales with the square of vapour pressure and may be guite weak in dry hot regions. This again makes me think that the atmospheric contribution from the dominate regions highlighted in fig 1 might be smaller than that of the surface. Hence, I am not convinced by the term 'large fraction' used in the statement on line 235. Given that the paper aims to provide insight into the mechanism behind the EOR signal means that this 'fraction' should be guantified. I therefore encourage the authors to perform a few simple RT runs, even using idealized atmospheres, so to make the attribution of the OLR signal clearer.

the top of the atmosphere when the surface temperature is increased. We also looked at the surface and 2-m temperature variations in the model. Using this information, we find that the magnitude of this pattern is actually dominated by changes in surface emission, rather than atmospheric emission.

• We found these calculations particularly insightful and have therefore decided to include them in the revised manuscript along with a new figure on L707-716, copied below.

[revised manuscript text omitted]

|                                                                                                                                                                                                                                                                                                                                                                                                                                                                                                                                                                                                                                                                                                                                                                                                                                                                                                                                                                                                                                             |  <li>winter atmosphere, or from the September 2010
minimum and maximum 2 m model temperatures
in the Sahara Desert, and is consistent with the fact
that the atmosphere is an order of magnitude less
efficient at increasing OLR for a given change in
temperature (Soden, 2008). We therefore conclude
that the first spatial-temporal pattern in the diurnal
cycle of OLR is dominated by increased surface
emission."</li> <li>Additional references used in this section "Soden,
B. J., Held, I. M., Colman, R., Shell, K. M., Kiehl,
J. T., and Shields C. A.: Quantifying Climate
Feedbacks Using Radiative Kernels. J. Climate, 21,
3504–3520, doi:10.1175/2007JCL12110.1, 2008."
and "Ogawa, K. and Schmugge T.: Mapping
Surface Broadband Emissivity of the Sahara Desert
Using ASTER and MODIS Data. Earth Interact., 8,
1–14, doi:10.1175/1087-
3562(2004)008<0001:MSBEOT>2.0.CO;2, 2004."
have been added to the reference list.</li> <li>The previous passage of text "Although it is
primarily the surface that is being heated, it should
be noted that transmittance of longwave radiation
back through the atmosphere is often low, typically
less than 10 % at the global scale (Costa and Shine,
2012). A large fraction of the variation in OLR
reaching the top of the atmosphere as a result of
solar heating of the land surface is therefore likely
to be due to radiation that has been absorbed and
re-emitted by the atmosphere." has been removed.</li> <li>Inserted "with the exception of some simplified
calculations for the dominant OLR pattern" on
L 388-389 for consistency with the above inclusion</li>  |
|---------------------------------------------------------------------------------------------------------------------------------------------------------------------------------------------------------------------------------------------------------------------------------------------------------------------------------------------------------------------------------------------------------------------------------------------------------------------------------------------------------------------------------------------------------------------------------------------------------------------------------------------------------------------------------------------------------------------------------------------------------------------------------------------------------------------------------------------------------------------------------------------------------------------------------------------------------------------------------------------------------------------------------------------|------------------------------------------------------------------------------------------------------------------------------------------------------------------------------------------------------------------------------------------------------------------------------------------------------------------------------------------------------------------------------------------------------------------------------------------------------------------------------------------------------------------------------------------------------------------------------------------------------------------------------------------------------------------------------------------------------------------------------------------------------------------------------------------------------------------------------------------------------------------------------------------------------------------------------------------------------------------------------------------------------------------------------------------------------------------------------------------------------------------------------------------------------------------------------------------------------------------------------------------------------------------------------------------------------------------------------------------------------------------------------------------------------------------------------------------------------------------------------------------------------------------------------------------------------------------------------------------------------------------------------------------------------------------------------------------------------------------------------------------------------------------------------------|
| Line 281: I generally like that the analysis (i.e. Fig 4) that the authors have performed on investigating the causes behind the 'U' shape. However, the impact of aerosol and mean cloudiness could be dealt with a bit better. Here the authors say they use only one aerosol case; 'rural aerosol' to see how aerosol loading could change the relationship between SZA and TOA albedo. Would it not be more useful to look at the extremes between say a highly scattering aerosol environment(e.g. high SO4 or sea salt) versus a highly absorbing aerosol environment (e.g black carbon). Just using one simple aerosol case does not really provide much insight into how much aerosol can alter the diurnal cycle of RSR. Also I wonder about impact of the mean state of cloudiness (as opposed to the diurnal cycle). I suspect that this 'U' shape would be stronger for cloudy versus clear regions (as shown in fig 5), but may get weaker as the mean cloudiness of a region goes up. That is because the amount of radiation |  <li>Your suggestion of including different aerosol types (scattering vs. absorbing) is something that we carefully considered pre-submission. The plot that we thought about including, that is similar to the previous Fig. 4b, is below. "rural" and "oceanic" aerosols are highly scattering whereas "urban" aerosols are highly absorbing. Optical properties are given in Ricchiazzi et al. (1998). AOD at 550 nm is 1 in all cases.</li> <li>O.6 Std. atm. + rural aerosol</li> <li>Std. atm. + urban aerosol</li> <li>Std. atm. + oceanic aerosol</li> <li>The provide this plot here for reference. The reason that we chose not to include this plot here for</li>                                                                                                                                                                                                                                                                                                                                                               |

| scattered to space per unit optical depth decreases with increasing cloud optical depth.                                                                                                                                                        | this plot in the main manuscript is that different
aerosol optical properties only provide a scaling of
the 'U' shape, and therefore do not change the
overall story, as pointed out on L314-316.                                                                                                                                                                                                                                                                          |  |  |  |
|-------------------------------------------------------------------------------------------------------------------------------------------------------------------------------------------------------------------------------------------------|-------------------------------------------------------------------------------------------------------------------------------------------------------------------------------------------------------------------------------------------------------------------------------------------------------------------------------------------------------------------------------------------------------------------------------------------------------------------------------------|--|--|--|
|                                                                                                                                                                                                                                                 | • This scaling relationship can also be shown
mathematically. For an optically thin atmosphere
over a black surface, the bidirectional reflection
distribution function (BRDF) of the system, $R_a$ , is
given by Liou (2002) as                                                                                                                                                                                                                                        |  |  |  |
|                                                                                                                                                                                                                                                 | $R_a(\mu,\phi;\mu_0,\phi_0) = \frac{\widetilde{\omega}\tau}{4\mu\mu_0}P(\mu,\phi;-\mu_0,\phi_0)$                                                                                                                                                                                                                                                                                                                                                                                    |  |  |  |
|                                                                                                                                                                                                                                                 | where $\mu$ and $\mu_0$ are the cosine of viewing and solar
zenith angles, respectively; $\phi$ and $\phi_0$ are viewing
and solar azimuth angles, respectively; $\tau$ and $\tilde{\omega}$ are
atmospheric optical depth and associated single
scattering albedo, respectively; and P is the phase
function. From this equation, we can see the clear
dependence of $R_a$ on $1/\mu_0$ , and the scaling of this
shape provided by $\tilde{\omega}$ . |  |  |  |
|                                                                                                                                                                                                                                                 | • Good point. It is true that the amount of radiation scattered to space per unit optical depth decreases with increasing cloud optical depth, although this is difficult to see in the EOF plots. We have added "Note that the U-shape can also become weaker as the mean cloudiness of a region increases, because the amount of radiation scattered to space per unit optical depth decreases with increasing cloud optical depth." on L332-334.                                 |  |  |  |
| Line 335: I would place more emphasis on this result
in the abstract and conclusions. The fact that your
technique of analyzing the diurnal cycle highlights
some clear limitations of the NWP cloud fields is an
important result. | • OK. Removed "to exist" and added "but with slightly different timings due to known model biases" on L24 in the abstract and added "The timing of the pattern related to cloud variations is slightly later in the observations, consistent with previous findings, but the presence of the patterns indicates" on L491-492 in the conclusions.                                                                                                                                    |  |  |  |

**Author updates**

| Manuscript change                                                           | Comment                                                                              |  |  |  |
|-----------------------------------------------------------------------------|--------------------------------------------------------------------------------------|--|--|--|
| Added ", and two anonymous reviewers for their thoughtful comments" on L521 | • Thank you for reviewer comments in the Acknowledgements                            |  |  |  |
| Updated figure numbers throughout                                           | • By adding a new figure (Fig. 2), all following figure numbers have increased by 1. |  |  |  |
| Moved "artificially" on L104                                                | • Improved wording                                                                   |  |  |  |

[revised manuscript text omitted]